# Inherently Robust Control through Maximum-Entropy Learning-Based Rollout

**Felix Bok**                                                                    *felix.bok@tum.de*
*Volkswagen Group & Technical University of Munich*
*Munich*

**Atanas Mirchev**
*Volkswagen Group*
*Munich*

**Barış Kayalıbay**
*Foundation Robotics Labs*
*Munich*

**Ole Jonas Wenzel**
*Foundation Robotics Labs*
*Munich*

**Patrick van der Smagt**
*Foundation Robotics Labs & ELTE University*
*Munich & Budapest*

**Justin Bayer**

**Reviewed on OpenReview:** *https://openreview.net/forum?id=Ho4XUDn21D*

## Abstract

Reinforcement Learning has recently proven extremely successful in the context of robot control. One of the major reasons is massively parallel simulation in conjunction with controlling for the so-called "sim to real" gap: training on a distribution of environments, which is assumed to contain the real one, is sufficient for finding neural policies that successfully transfer from computer simulations to real robots. Often, this is accompanied by a layer of system identification during deployment to close the gap further. Still, the efficacy of these approaches hinges on reasonable simulation capabilities with an adequately rich task distribution containing the real environment. This work aims to provide a complementary solution in cases where the aforementioned criteria may prove challenging to satisfy. We combine two approaches, *maximum-entropy reinforcement learning* (MaxEntRL) and *rollout*, into an inherently robust control method called **Maximum-Entropy Learning-Based Rollout (MELRO)**. Both promise increased robustness and adaptability on their own. While MaxEntRL has been shown to be an adversarially-robust approach in disguise, rollout greatly improves over parametric models through an implicit Newton step on a model of the environment. We find that our approach works excellently in the vast majority of cases on both the Real World Reinforcement Learning (RWRL) benchmark and on our own environment perturbations of the popular DeepMind Control (DMC) suite, which move beyond simple parametric noise. We also show its success in "sim to real" transfer with the Franka Panda robot arm.

# 1    Introduction

Reinforcement Learning (RL) identifies optimal behavior strategies through an automated process of trial and error and constitutes one of today's standard approaches for addressing complex sequential decision-making problems. In particular, the application of RL to humanoid robots has achieved extremely impressive results. For example, they can now successfully navigate through complex terrain (Sun et al., 2025), dexterously manipulate real-world objects purely from vision (Lin et al., 2025), and even perform complex dynamic movements (Zhuang et al., 2024). In most cases, learning policies for such tasks rely on massively parallel simulations, given that training directly on hardware is impractical in nearly all instances: often, millions of interaction steps are required. However, training control policies exclusively in simulation frequently uncovers a phenomenon referred to as the "sim to real" gap: a policy that has been extensively optimized in simulation demonstrates degraded performance, or even complete failure, during real-world deployment.

One successful and commonly used approach is domain randomization (Tobin et al., 2017), where the policy is exposed to a wide range of simulated environments. Key simulation parameters are randomized during policy training, creating more diverse experiences and forcing policies to become robust to environmental variations. The real world is then assumed to be just another variation within that distribution. Fulfilling this assumption rests on two key desiderata: (i) all critical environment parameters and their ranges need to be carefully identified such that the real world is close or contained within; (ii) the space needs to be sufficiently tight to avoid overly conservative policies that need to compromise performance over too many different instances. Whereas the last point can be addressed in parts by domain identification, this process requires precise domain knowledge, extra engineering effort, and additional computing power to work well (Josifovski et al., 2022; 2024; Dulac-Arnold et al., 2021).

In this paper, we present an orthogonal approach to domain randomization tailored to instances where it is challenging to specify all key simulation parameters or too costly to exhaust all. To that end, we integrate *maximum-entropy reinforcement learning* (MaxEntRL) (Ziebart et al., 2008) and *rollout* (Tesauro, 1994; Bertsekas, 2024) into an inherently robust method called **M**aximum-**E**ntropy **L**earning-Based **Ro**llout (**MELRO**). While MaxEntRL enhances policy robustness by promoting exploratory behaviors, rollout compensates for residual uncertainties through real-time re-planning. We demonstrate in extensive evaluations that MELRO features and extends the strengths of both frameworks. Firstly, we conduct multiple experiments on an illustrative toy example that aim to highlight the differences between the strategies generated by traditional MaxEntRL, domain randomization and MELRO. These experiments also provide in-depth insights into the effects of entropy-regularization planning. Secondly, we benchmark its performance against a variety of tasks in the Real World Reinforcement Learning (RWRL) suite. Furthermore, we assess its capabilities on a custom set of challenging environment perturbations applied to the popular DeepMind Control (DMC) suite, which are designed to probe robustness beyond simple parametric noise. Finally, we present a successful "sim to real" transfer of MELRO on the Franka Panda robot arm.

# 2    Preliminaries

## 2.1    Problem

We assume that the problem consists of an unknown number of Markov Decision Processes (MDPs) (Bellmann, 1957) sharing the same action and state space. This can be formalized as a partially observable Markov Decision Process (POMDP) (Kaelbling et al., 1998), which is characterized by the tuple $(\mathcal{S}, \mathcal{A}, \Omega, p, r, o, p_0, \gamma)$, where $\mathcal{S}$, $\mathcal{A}$ and $\Omega$ are the state, action and observation spaces, $p : \mathcal{S} \times \mathcal{A} \mapsto \mathcal{S}$ is the transition function, $r : \mathcal{S} \times \mathcal{A} \mapsto \mathbb{R}$ is the reward function, $o : \mathcal{S} \mapsto \Omega$ is the observation function, $\gamma \in [0, 1]$ is a discount factor, and $p_0$ is the initial state distribution (Humplik et al., 2019). The state space $\mathcal{S}$ then consists of the state space of the MDPs, denoted as $S^{\mathcal{M}}$, and a set of task distributions $\mathcal{T}$. Initially, the task specification $\tau$ is sampled from $\mathcal{T}$ and maintained through time, assuming it is the only unobservable part of the state. This results in $S = (S^{\mathcal{M}} \times \mathcal{T})$, $\Omega = S^{\mathcal{M}}$ and $o((s, \tau)) \mapsto s$, where $s \in S^{\mathcal{M}}$. The transition and reward functions and the initial state distribution are also conditioned on $\tau$. The goal is then to find a policy $\pi : S^{\mathcal{M}} \mapsto \mathcal{A}$

that maximizes the expected discounted return under the sampled but unknown task specification $\tau$:

$$\pi(a_k|s_k) = \arg\max_{a_k} \max_{a_{k+1:T}} \mathbb{E}_{\Gamma^\tau} \left[ \sum_{t=k}^{T} \gamma^t r(s_t, \tau, a_t) \right], \tag{1}$$

where $\Gamma^\tau = s_{k:T}$ is a state trajectory where the states are sampled from the transition function of the MDP $\tau$, and $k \leq T \in \mathbb{N}$.

## 2.2 Maximum-Entropy Reinforcement Learning

Maximum-entropy reinforcement learning (MaxEntRL) extends traditional RL by augmenting Equation 1 with a conditional regularization term, the policy entropy $\mathcal{H}_\pi$:

$$\pi_{\text{MaxEnt}}(a_k|s_k) = \arg\max_{a_k} \max_{a_{k+1:T}} \mathbb{E}_{\Gamma^\tau} \left[ \sum_{t=k}^{T} \gamma^t r(s_t, \tau, a_t) \right] + \eta \mathcal{H}_\pi(a_k|s_k),$$
$$\mathcal{H}_\pi(a_k|s_k) = \mathbb{E}_{a \sim \pi(a_k|s_k)} \left[ -\log \pi(a|s_k) \right], \tag{2}$$

where $\eta > 0$ is a temperature parameter controlling the entropy trade-off and the entropy $\mathcal{H}_\pi(a_k|s_k)$ is approximated with Monte Carlo integration. Solving for Equation 2 results in stochastic policies characterized by non-zero probability for every action at each state. The entropy is higher when more actions lead to similar rewards and lower when one action is substantially better.

## 2.3 Learning-Based Rollout

Rollout[1] is a control method employing a dynamic system model to predict future behavior and optimize controls online. An elegant theoretical interpretation of rollout positions it as implementing a Newton step in value space, a perspective described by Bertsekas (2024). At each time step, rollout optimizes the controls for a given lookahead horizon $L \in \mathbb{N}$ to minimize the predicted future costs and applies the first optimized control in a receding-horizon fashion:

$$\pi_{\text{r}}(a_k|s_k) = \arg\max_{a_k} \max_{a_{k+1:k+L-1}} \mathbb{E}_{\Gamma^\tau} \left[ r(s_k, \tau, a_k) + \sum_{t=k+1}^{k+L-1} \gamma^{t-k} r(s_t, \tau, a_t) + \gamma^L R(s_{k+L}) \right], \tag{3}$$

where $R : \mathcal{S}^\mathcal{M} \mapsto \mathbb{R}^+$ is the terminal reward function estimating all future rewards and $\Gamma^\tau = s_{k:k+L}$. The rollout components, the dynamics model, reward, and terminal cost function are typically hand-engineered. Instead, we use multi-layer perceptrons (MLPs) whose parameters are learned from environmental interactions.

# 3 Maximum-Entropy Learning-Based Rollout

MELRO uses the control framework introduced by Bertsekas (2022). It consists of two stages: *off-line training* and *on-line play*. The approach is visualized in Figure 1.

## 3.1 Off-line Training

In *off-line training*, we use a model-based reinforcement learning (MBRL) framework to train a model, a base policy, and a critic.

**Model Learning**  It is important to note that MELRO is model-agnostic; consequently, it can be utilised in conjunction with any prediction model. In this study, the model architecture and training are based on the variational state-space model utilised in Bayer et al. (2021), with the following modifications. Given that

---

[1]Other related terms are model predictive control (MPC), receding horizon control, and limited lookahead control. We will use the term "rollout", as in the work of Bertsekas (2024).

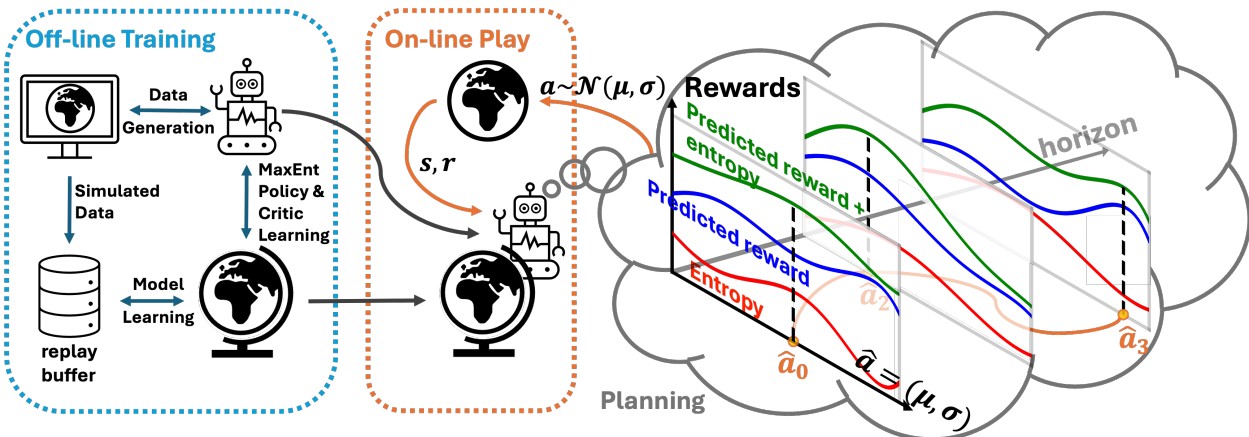

Figure 1: **Overview of MELRO.** We propose a framework that uses both maximum-entropy reinforcement learning and rollout to achieve inherently robust control complementary to domain randomization. Our method incorporates maximum-entropy regularization during off-line training and on-line planning. To achieve this, the world model is trained on observations and action distribution parameters. Planning then occurs directly within the action distribution parameter space.

each individual task considered here is fully observable, the emission model directly propagates the states forward as the mean of a Gaussian with fixed variance:

$$p(x_t|s_t) = \mathcal{N}\left(x_t|\mu = s_t, \sigma_x^2 = 0.01\right). \tag{4}$$

Moreover, a cost-function head is introduced, which maps states to rewards:

$$r_{\theta_r}(r_t|s_t, \hat{a}_t) = \text{MLP}_{\theta_r}(s_t, \hat{a}_t), \tag{5}$$

where $\hat{a} = (\mu, \sigma)$ is the mean and the standard deviation of a Gaussian distribution. The transition is modeled as a Gaussian residual component where the output is a convex combination of the previous state $s_{t-1}$ and an update $\hat{s}_{t-1}$. The mixing coefficient $\rho$ and the update are outputs of a MLP with parameters $\theta_s$:

$$p_{\theta_s}(s_t|s_{t-1}, \hat{a}_{t-1}) = \mathcal{N}\left(\mu = \text{RC}_{\theta_s}(s_{t-1}, \hat{a}_{t-1}), \sigma_s^2 = 0.001\right) \tag{6}$$

$$\text{RC}_{\theta_s}(s_{t-1}, \hat{a}_{t-1}) = \rho * s_{t-1} + (1 - \rho) * \hat{s}_{t-1} \tag{7}$$

$$(\hat{\rho}, \hat{s}_{t-1}) = \text{MLP}_{\theta_s}(s_{t-1}, \hat{a}_{t-1}), \quad \rho = \frac{1}{2}(\texttt{softsign}(\hat{\rho}) + 1). \tag{8}$$

This enables states to propagate more easily through transitions. All model components are trained on a replay buffer containing all previous policy-environment interactions.

**Policy Learning** The policy is modeled as a normal distribution, parametrized by $\phi$, where a two-headed MLP estimates the mean and standard deviation:

$$\pi_\phi(a_t|s_t) = \mathcal{N}\left(a_t|(\mu_t, \sigma_t) = \text{MLP}_\phi(s_t)\right). \tag{9}$$

The critic $\hat{v}_\chi$ is modeled as a deterministic MLP with parameters $\chi$. Both the policy and the critic use layer normalization (Ba et al., 2016). The critic loss $\mathcal{L}_{\hat{v}_\chi}$ is given by a supervised learning objective on TD($\lambda$)-targets which are computed on imagined model and policy rollouts:

$$\mathcal{L}_{\hat{v}_\chi} = \mathbb{E}_{a_t \sim \pi_\phi}\left[\frac{1}{H_v}\sum_{t=0}^{H_v}\frac{1}{2}\|\hat{v}_\chi(s_t, a_t) - R(s_t)\|_2^2\right], \tag{10}$$

with TD($\lambda$)-targets:

$$R(s_t) = r_{\theta_r}(s_t, a_t) + \eta \mathcal{H}_{\pi_\phi}(a_t|s_t) + \gamma \left((1 - \lambda)\texttt{sg}\left(\hat{v}_\chi(s_t, a_t)\right) + \lambda R(s_{t+1})\right), \tag{11}$$

$$R(s_{t+H_v}) = \texttt{sg}\left(\hat{v}_\chi(s_{t+H_v}, a_{t+H_v})\right), \tag{12}$$

where $\texttt{sg}$ is the $\texttt{stop-gradient}$ operator. The policy loss $\mathcal{L}_{\hat{\pi}_\phi}$ is given by the expected values of the TD($\lambda$)-targets:

$$\mathcal{L}_{\pi_\phi} = \mathbb{E}_{a_t \sim \pi_\phi}\left[\sum_{t=0}^{H_\pi} R(s_t)\right]. \tag{13}$$

The optimal parameters for the policy and critic are obtained by maximizing the weighted sum of the policy and critic loss on model rollouts. Additionally, a regularization term consisting of the norm of the gradients of the policy and critic loss is introduced to improve the convergence of the optimization problem by increasing the conservativity (Mescheder et al., 2017). The policy-critic loss is then given by:

$$\mathcal{L}_{\pi_\phi, \hat{v}_\chi} = \zeta \mathcal{L}_{\pi_\phi} + \mathcal{L}_{\hat{v}_\chi} + \eta * \left(\frac{\left\|\nabla \mathcal{L}_{\pi_\phi}\right\|_2^2}{\left|\nabla \mathcal{L}_{\pi_\phi}\right|} + \frac{\left\|\nabla \mathcal{L}_{\hat{v}_\chi}\right\|_2^2}{\left|\nabla \mathcal{L}_{\hat{v}_\chi}\right|}\right), \tag{14}$$

where the loss is scaled by a weighting hyperparameter $\zeta$ and the conservativity regularization term by $\eta$. During training, the gradients of all components are clipped by the Euclidean norm, with a bound specified for each component.

### 3.2 On-line Play

In *on-line play*, we combine the trained components into the rollout algorithm. We thereby improve standard rollout in the following ways. First, online planning happens in the space of action means and standard deviations of a Gaussian policy. Secondly, we generate imagined rollouts for the planning horizon based on the learned world model and the base policy. The resulting base policy action distribution parameters, i.e. means and standard deviations over time, are utilized as initial values for gradient descent. Thirdly, in an analogous manner to standard MaxEntRL, the rollout cost function, as defined in Equation 3, is regularized by the entropy of the action distribution. The idea behind this regularization is that it will lead to more robust actions being selected during planning. Furthermore, immediately following the model rollouts and before applying the critic, a fixed number of base policy steps are performed, as suggested by Bertsekas (2024). This truncated rollout with the base policy enhances the stability property of the rollout policy at a low additional computational cost.

The resulting rollout problem is then given by the following equations:

$$\pi_{\text{MELRO}}(\hat{a}_k|s_k) = \arg \max_{\hat{a}_k} \max_{\hat{a}_{k+1}, \dots, \hat{a}_{k+L}, \hat{a}_{k+L+m}} \mathcal{L}_{\text{MELRO}},$$

$$\mathcal{L}_{\text{MELRO}} = \mathbb{E}\left[r_{\theta_r}(s_k, a_k) + \sum_{i=k+1}^{k+L-1} \gamma^{i-k} r_{\theta_r}(s_i, a_i) + \tilde{R}_{k+L}\right] + \eta \sum_{j=k}^{k+L-1} \mathcal{H}_\pi(\hat{a}_j|s_j), \tag{15}$$

$$\tilde{R}_{k+L} = \sum_{j=k+L}^{k+L+m-1} r_{\theta_r}(s_j, a_j) + \gamma^{k+L+m} \hat{v}_\chi(s_{k+L+m}, a_{k+L+m}),$$

where $\hat{a}_i = (\mu_i, \sigma_i)$, $a_i \sim \mathcal{N}(\mu_i, \sigma_i)$ $\forall i \in \{k, k+1, \dots, k+L, a_k+L+m\}$, $a_{k+L:k+L+m-1}$ is sampled from the base policy $\pi_\phi$, $s_{k:k+L+m}$ is sampled from the transition function $p_{\theta_s}$ and $\tilde{R}_{k+L}$ is the approximation of the future rewards starting from the time step $k + L$. The final controls are then computed in a receding-horizon fashion, optimizing Equation 15 using Adam (Kingma & Ba, 2015), where the search direction in the space of action means and standard deviations is generated using Augmented Random Search (ARS) (Mania et al., 2018).

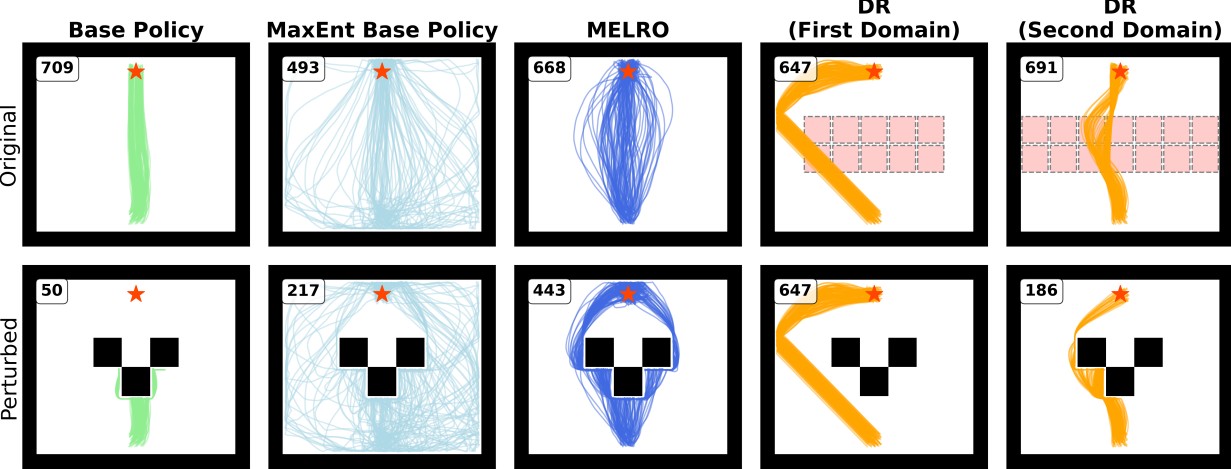

Figure 2: **Results for the toy maze.** 100 evaluation trajectories in the unperturbed training maze and the perturbed evaluation maze of the no entropy base policy, MaxEnt base policy, MELRO, and two policies trained using domain randomization with different domains. The mean reward for each method and environment is shown in the top left corner of every maze. The goal is visualized by the red star. The red blocks in the original environment of the two policies trained using domain randomization indicate where an individual block can appear during training in a rollout with a probability of 0.5. During the evaluation of the original environment, no block was in the maze.

## 4 Experiments

To evaluate the robustness and transfer capabilities, experiments were conducted for an illustrative toy example, in simulation, and on a real Franka Panda robot arm. Training and validation for all experiments have been conducted on a MIG 1g.10gb partition of an NVIDIA A100 GPU (compare to NVIDIA (2025) for more details). An evaluation of the computational costs is provided in Appendix C.

### 4.1 Illustrative Toy Example

To illustrate the challenges posed by traditional MaxEntRL methods and domain randomization, and to highlight the advantages of MELRO, we use a simple 2-DoF point maze environment modeled with the Gymnasium-Robotics suite (de Lazcano et al., 2024). The task in this environment is to move a force-actuated ball from the bottom middle to the top middle of a closed maze, with the exponential negative Euclidean distance between the current position and the goal position as reward. In order to simulate perturbations and the "sim to real" gap, the training of all components is performed without the incorporation of any blocks in the middle of the maze. The evaluation is then conducted on a maze that contains three blocks directly positioned between the starting position and the goal. To approximate the optimal level of entropy regularization, we use the perturbed environment to validate multiple MaxEnt base policies with different entropy weights.

**Challenges of traditional MaxEntRL** The three left columns of Figure 2 present 100 trajectories and their mean rewards for the base policy without entropy regularization, the MaxEnt base policy, and MELRO. The results highlight that policies with no or insufficient entropy regularization tend to overfit to the unperturbed environment and fail completely once perturbations are introduced, as all trajectories become stuck at the blocks. When the entropy weight is tuned appropriately, MaxEntRL generates more entropic and diverse trajectories, improving robustness and enabling more trajectories to reach the goal. However, performance in the unperturbed environment deteriorates significantly, and identifying the appropriate regularization level is computationally costly, as a change of the entropy weight necessitates a complete MBRL

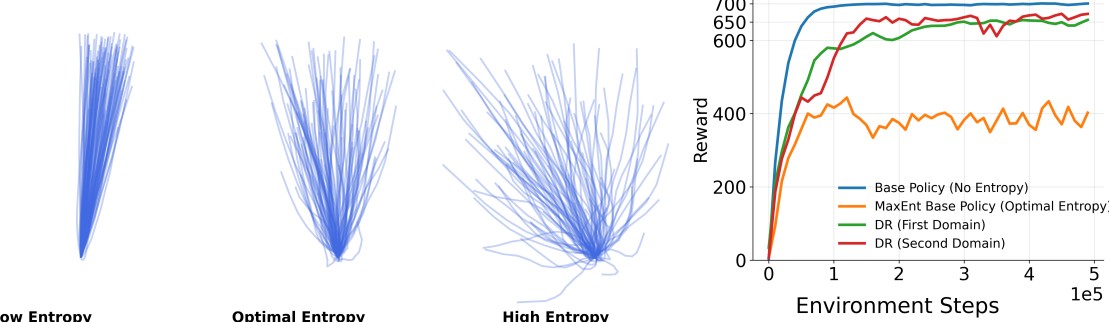

Figure 3: **Effects of the entropy weight on the rollout plans.** Visualization of rollout plans of MELRO with different entropy regularization during planning on the toy maze example.

Figure 4: **Training curves toy example.** Reward during training for all the methods evaluated for the toy maze example.

run. Moreover, high entropy regularization results in highly stochastic trajectories that can degrade policy learning, as shown in Figure 4.

**Advantages of MELRO** In contrast, MELRO consistently achieves the optimal balance: it relinquishes only a minimal amount of reward in the unperturbed environment and attains the maximum mean reward under the perturbation. One reason for this is its capability for adapting the entropy regularization during the on-line play phase for the specific perturbation. This makes tuning the regularization far more efficient. Furthermore, adjusting the entropy weight within MELRO has an intuitive and interpretable effect of smoothly broadening the diversity of the paths taken. This is the direct effect of entropy regularization increasing the standard deviation of the action distribution. Consequently, this results in a greater degree of diversification in the plans during the planning process, as illustrated in Figure 3. Lastly, given that the components of the best-performing MELRO configuration are trained in a low-entropy regularized MBRL run, the trajectories are more directed towards the goal. The reason for this is that MELRO can maintain dynamic entropy adaptation in a manner analogous to the low-entropy base policy. Further evaluations of MELRO can be found in Appendix G.1.

**Comparison of MELRO to Domain Randomization** To compare MELRO with domain randomization, two policies have been trained under two randomized domains. The domains differ in regions where blocks can appear, during training and for each single rollout, with a probability of 0.5. These and the resulting trajectories are depicted in the most right columns of Figure 2. The policy trained on the first domain converged to a conservative strategy that nearly always avoids the area where blocks can appear, resulting in identical trajectories across mazes. While this approach yields the highest mean reward in the perturbed environment, it slightly underperforms MELRO in the unperturbed setting and risks failure if the sole traversed path becomes blocked. Conversely, the policy trained on the second domain, where every path to the goal may be obstructed, learns to follow approximately the shortest route, effectively disregarding perturbations. This strategy achieves slightly higher performance on the unperturbed maze, but exhibits a significantly lower level of robustness when compared with trajectories of MELRO on the perturbed maze.

These results highlight the sensitivity of domain randomization to the chosen simulation parameter distribution: if the randomized domain is too narrow, resulting policies may become overly conservative or insufficiently robust. Moreover, the comparison underscores that domain randomization and maximum-entropy regularization induce robustness through complementary and orthogonal mechanisms.

## 4.2 Simulation

We evaluated MELRO across four simulated environments from the DMC suite (Tassa et al., 2018), paired with five distinct perturbations. Four perturbation types - torso and thigh length for "Walker Walk", and shin length and joint damping for "Quadruped Walk" - have been adapted from the RWRL suite

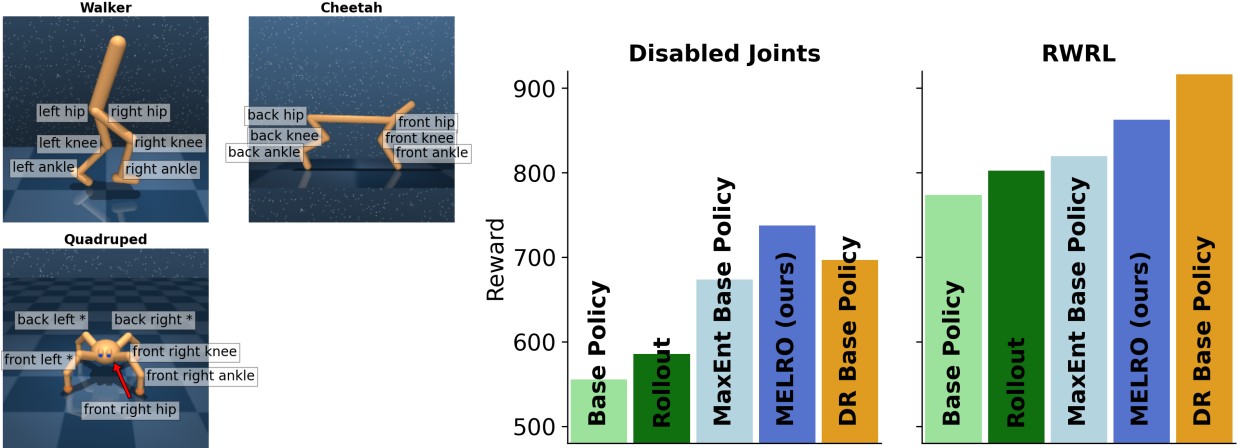

Figure 5: **Disabled joints tasks.** Visualization of the Walker, Cheetah, and Quadruped joints, which have been individually disabled.

Figure 6: **Results of all simluation experiments.** Average reward of all evaluated methods over all simulated environments and perturbations.

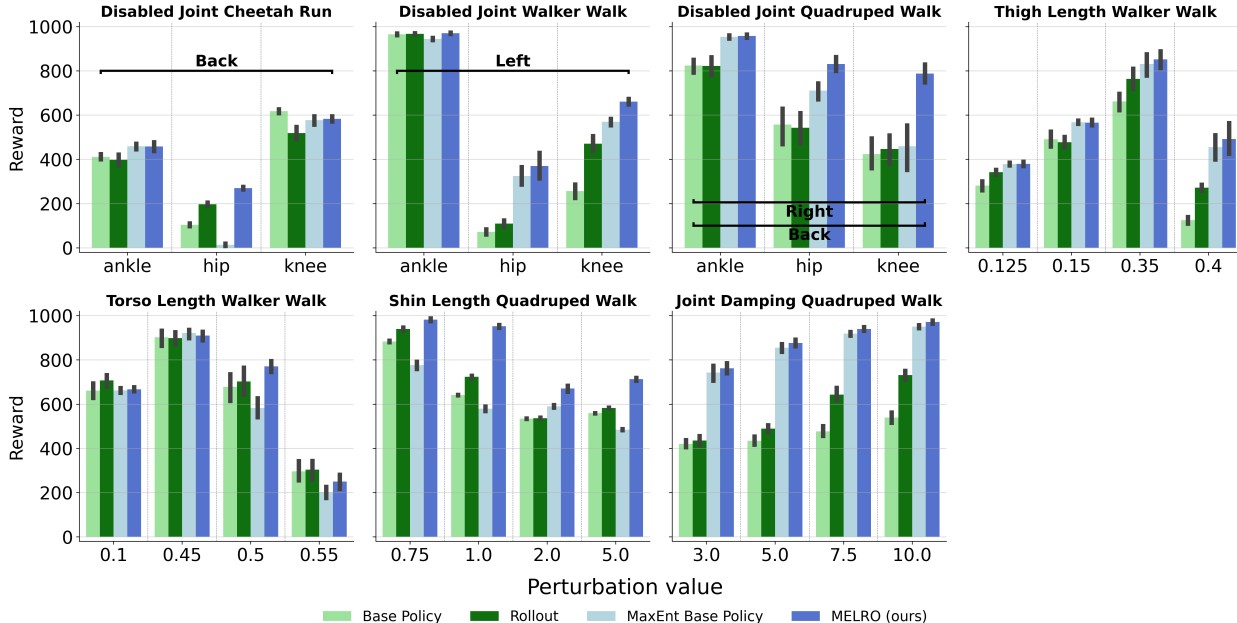

Figure 7: **Individual results of all simulation experiments.** Average reward and 95%-CI of the non-MaxEnt base policy, rollout, MaxEnt base policy, and MELRO for a subset of perturbation tasks from the RWRL and the DMC suite. The horizontal brackets indicate the positioning of the joints.

(Dulac-Arnold et al., 2020). The fifth perturbation type – disabled joints for "Walker Walk", "Quadruped Walk", and "Cheetah Run" – has been extended on Nagabandi et al. (2019). Even though the joints repeat for each extremity, we still observed differences in the performance of all evaluated methods, so we disabled the joints individually. This perturbation type introduces strong non-parametric disturbances, which move beyond simple noise. The joints for each environment are visualized in Figure 5.

We ablate MELRO against its MaxEnt base policy, the base policy trained without MaxEntRL, and standard rollout over the latter. Despite the evident advantage of policies trained under domain randomization in terms of the opportunity to encounter the specific validation perturbations during the MBRL phase, an additional

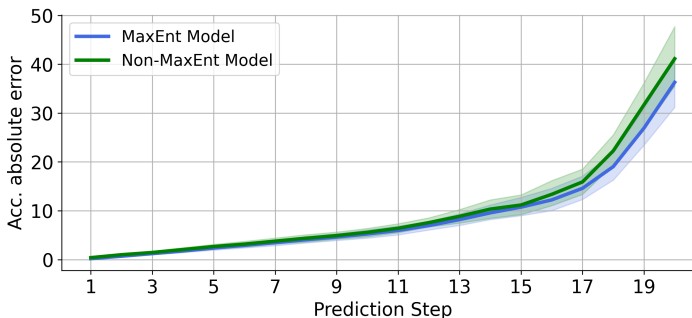
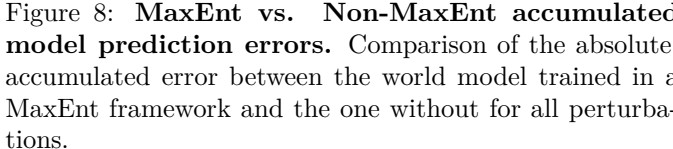

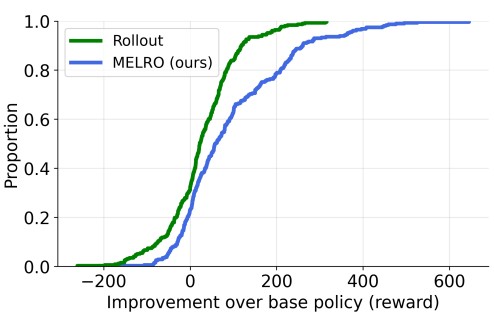

Figure 8: **MaxEnt vs. Non-MaxEnt accumulated model prediction errors.** Comparison of the absolute, accumulated error between the world model trained in a MaxEnt framework and the one without for all perturbations.

Figure 9: **MELRO vs. plain rollout improvements.** ECDFs of reward improvements of MELRO and standard rollout relative to the respective base policy. More mass on the right translates to larger improvements.

comparison is made between MELRO and policies trained using domain randomization. The MaxEnt base policy, the base policy trained without entropy regularization, and their corresponding dynamics models were initially trained on standard DMC environments. We then picked the best-performing model, policy, and critic combination and optimized the rollout parameters for each perturbation through hyperparameter search. An evaluation of the hyperparameters and a sensitivity analysis can be found in Appendix B and D. The base policies using domain randomization have been trained directly on the respective perturbed simulations. During training, the specific disabled joint or perturbation value was randomly sampled from the entire domain. Final performance metrics were obtained by assessing each methodology across all perturbation tasks through ten independent random seed evaluations, each comprising ten independent rollouts.

**Main Simulation Results** The overall aggregated metrics are visualized in Figure 6. The results for a subset of perturbations of the entropy-regularized methods are shown in Figure 7, and of the domain randomization policies are shown in Figure 11 (Figures presenting all the perturbations are in Appendix G.2). In general, the MaxEnt policies produce superior results to the non-MaxEnt policies, and rollout enhances the performance of the respective base policy. This results in MELRO achieving the best overall performance compared with the non-regularized and entropy-regularized methods. MELRO even outperforms the base policy trained using domain randomization in the disabled joint task.

**Increased Inherent Robustness of MELRO** The MaxEnt base policy has already demonstrated robust performance across a broad spectrum of perturbations, achieving consistently high returns relative to the perturbation, in particular for all "Quadruped Walk" environments and the "Walker Walk" thigh length perturbations. Nevertheless, there were still some perturbations where the MaxEnt base policy exhibited significant degeneration in performance, often underperforming the non-regularized base policy in those cases. This phenomenon is especially apparent in the disabled back hip joint perturbation of "Cheetah Run", the shin length perturbation of 1.0 for "Quadruped Walk", and the torso length perturbation of 0.5 for "Walker Walk", as shown in Figure 7. In the disabled back hip joint perturbation of "Cheetah Run", the MaxEnt base policy even gets completely stuck at the beginning of all evaluated episodes. By contrast, MELRO consistently outperformed the MaxEnt base policy across all tested perturbation scenarios, particularly in cases where the MaxEnt base policy's performance deteriorated significantly, thereby demonstrating an enhanced dynamic entropy adaptation capability.

**Effects of Maximum-Entropy Regularization on World Models and Rollout** The training of a world model using MaxEntRL has already been demonstrated to enhance exploration, consequently leading to a more accurate world model (Ma et al., 2022). As demonstrated in Figure 8, this approach also results in a more robust model and lower prediction errors. Additionally, the superior model yields superior control

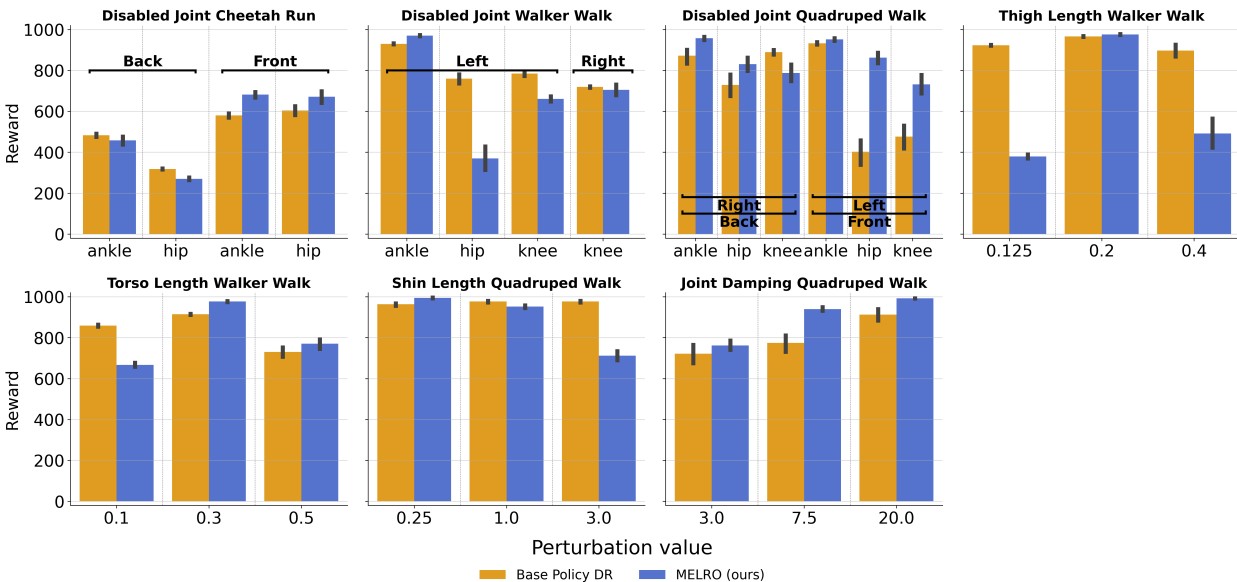

Figure 11: **MELRO vs. domain randomization perturbed simulation results.** Average reward and 95%-CI of domain randomization and MELRO for a subset of perturbation tasks from the RWRL and the DMC suite. The horizontal brackets indicate the positioning of the joints.

results. This is apparent from the more pronounced improvement of MELRO over its base policy, compared to the improvements shown by standard rollout (refer to Figure 9). A further explanation for this observation could be provided by Ahmed et al. (2019), in which the authors demonstrate that maximum-entropy regulation smooths the optimization landscape. This would also benefit the online optimization of rollout in MELRO.

**Limitations of Maximum-Entropy Regularization** Although maximum-entropy regularization has been demonstrated to improve robustness and exploration by introducing adaptive stochasticity in the action space, its effectiveness hinges on the environment (Ahmed et al., 2019; Eysenbach & Levine, 2022). In environments that require precise control, or where the perturbations differ just slightly from the training environment, maximum-entropy regularization can negatively impact performance. In our simulation experiments, this is evident in the "Cheetah Run" disabled back knee and front ankle, as well as some "Walker Walk" tasks, where both the MaxEnt base policy and MELRO perform slightly worse than the standard base policy.

**Comparison to Domain Randomization** As shown in the aggregated mean rewards in Figure 6, domain randomization achieves the best performance under the RWRL perturbation tasks, whereas MELRO outperforms it in the disabled joint settings. Overall, the individual results in Figure 11 suggest a general trend: Domain randomization is more effective in handling extreme perturbations, while MELRO provides superior robustness in scenarios where the perturbations have a smaller impact on the dynamics. A notable exception is the disabled joint "Quadruped Walk" task, where MELRO demonstrates significantly higher robustness, particularly when

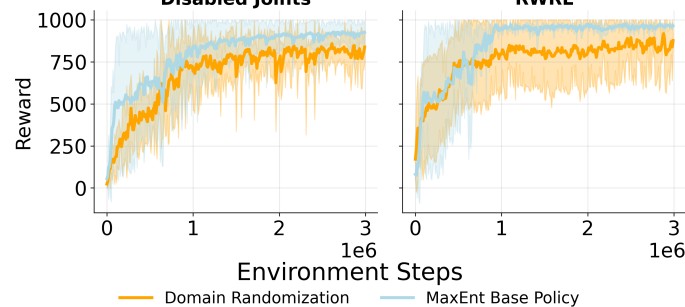

Figure 10: **MaxEntRL vs. domain randomization training curves.** Mean and 95%-CI reward training curves for the base policies trained using MaxEntRL and domain randomization aggregated across all evaluated tasks.

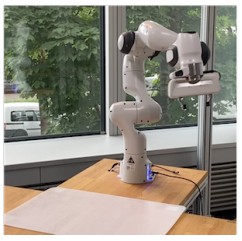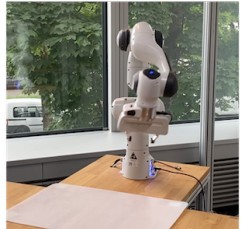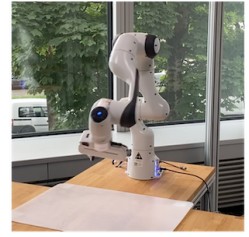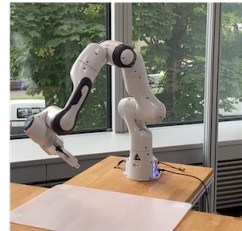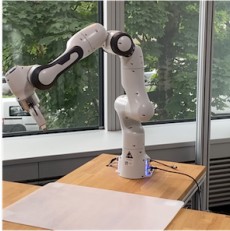

Figure 12: **Real-world Franka trajectory.** Example of a real-world trajectory of the Franka Panda robot arm, controlled by MELRO. The movement took approximately five seconds.

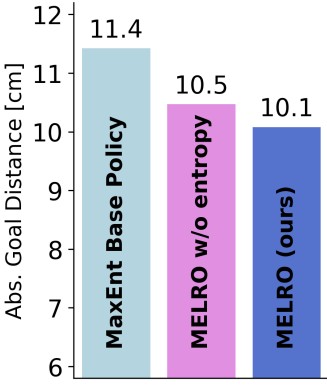

Figure 13: **Overview of the "sim to real" results for the Franka Panda reach task.** Mean of the absolute goal distances in cm of the MaxEnt base policy, MELRO without entropy regularization during planning, and MELRO on the real-world Franka Panda.

Figure 14: **MaxEnt base policy vs. MELRO individual differences after the "sim to real" transfer to the Franka Panda.** Difference in cm between the absolute distances between MELRO and the target, and between the MaxEnt base policy and the target, for all test runs on the real Franka Panda arm. Values above zero mean that MELRO was closer to the target than the MaxEnt base policy. Vice versa for values below zero.

joints from the two front legs are disabled. This phenomenon can potentially be attributed to the extensive variability induced by domain randomization in this task, forcing the policy to compromise across a wide range of possible dynamics. The reward training curves in Figure 10 indicate that training using domain randomization requires more environment interactions until the policy converges, and is, in general, noisier. However, this must be taken with a caveat, as the reward for the domain randomized policy is computed in perturbed environments.

### 4.3 Franka Panda Robot Arm

We evaluated MELRO's "sim to real" capabilities by testing it on a simple reach task using the Franka Emika Panda robot arm. The objective was to maneuver the end-effector at the end of the robot arm to a randomly selected position in front of the robot. To train the components in simulation, we modeled the task in MuJoCo (Todorov et al., 2012) using the Franka Panda MuJoCo Menagerie model (Zakka et al., 2022). The reward function has been selected as the negative sum of squares of the difference between the current end-effector position and the goal, minus a scaled penalty for rotating the end-effector out of a neutral orientation. It is important to note that no further reward shaping has been applied. Here, all policies – the base policy, rollout, MaxEnt base policy, and MELRO – have been trained to generate joint velocity actions given the current joint angles, joint velocities, and the current goal. Detailed information about the simulation is given in the Appendix E. The same training process as explained in Chapter 4.2

has been used. During deployment, all policies generated new joint velocity actions at a frequency of 10 Hz. These were then converted into joint torque commands using a PD-controller running at 1 kHz.

Although we evaluated multiple sets of hyperparameters for both the base policy and rollout, none of them produced stable and safe actions on the real robot. Since we did not apply extensive reward shaping, we believe that exploration for the non-MaxEnt components during the MBRL stage was insufficient. Consequently, MELRO was only evaluated against the MaxEnt base policy and MELRO without entropy regularization during planning. The mean distance for each method over the 20 test runs on the real Franka Panda is shown in Figure 13. Additionally, Figure 14 visualizes the differences in the absolute distances between MELRO and the target, and between the MaxEnt base policy and the target, for each test trajectory individually.

**MELRO tightens the "sim to real" gap** On average, MELRO was 11.7% closer to the goal than the MaxEnt base policy. This was achieved through faster convergence and generally bringing the end-effector closer to the goal. As can be seen from MELRO's good performance without entropy regularization, rollout already significantly improves the MaxEnt base policy. Additionally, the introduction of maximum-entropy regularization during planning further decreased the overall distance to the goal.

## 5 Related Work

### 5.1 Maximum-Entropy RL

It is well-known that MaxEntRL improves exploration (Haarnoja et al., 2017; Huang et al., 2020). This approach is also widely adopted, underpinning prominent model-free algorithms like SAC (Haarnoja et al., 2018), influencing implementations of methods like PPO (Schulman et al., 2017), and featuring in model-based approaches such as DreamerV3 (Hafner et al., 2023) and TD-MPC2 (Hansen et al., 2024). The primary distinction between the usage of MaxEntRL in MELRO and the aforementioned algorithms is that MELRO incorporates maximum-entropy regularization in the online planning objective.

Recent theoretical work has established that MaxEntRL implicitly optimizes a lower bound on a robust RL objective, conferring inherent resilience against perturbations in dynamics and rewards without the need for specialized robust optimization techniques (Eysenbach & Levine, 2022; Brekelmans et al., 2022). Despite this advantage, the resulting policies can sometimes be overly conservative, potentially sacrificing peak performance (Josifovski et al., 2022). Furthermore, standard MaxEntRL lacks explicit mechanisms for targeted adaptation, potentially hindering swift responses to rapidly evolving environments. Our work aims to address these limitations by harnessing the benefits of MaxEntRL while enabling more controlled robustness and faster adaptation by adding maximum-entropy regularized online planning.

### 5.2 Robustness in Model-Based Reinforcement Learning

In model-based reinforcement learning (MBRL), model inaccuracies are commonly addressed by incorporating model uncertainty into the learning objective (Janner et al., 2019; Wang et al., 2024) or directly into the decision-making process (Chua et al., 2018; Wu et al., 2022). Epistemic uncertainty is typically estimated by ensemble disagreement, but can also be predicted directly by the model. Although this method encourages the selection of actions that are more predictable and therefore likely to be more robust, it also leads to less exploration and, consequently, to suboptimal policies. Additionally, all methods rely on accurate estimates of uncertainty. Particularly for longer planning horizons, reduced performance and sampling efficiency have been observed as the inevitable model inaccuracies are interpreted as model uncertainty (Wang et al., 2024). Given that MELRO employs MaxEnt regularization, it is not subject to these limitations.

## 6 Conclusion

We introduced MELRO, a novel, inherently robust control method that combines maximum-entropy reinforcement learning (Haarnoja et al., 2018) with rollout, an online planning method (Tesauro, 1994). Instead of relying on extensive domain randomization to bridge the "sim to real" gap, we explored methods that are

inherently more robust to such deviations from the model at hand. It is hence able to cope with variations that are unknown a priori. Experimentally, this approach significantly enhances system robustness against diverse and unforeseen dynamical perturbations, as well as the success of "sim to real" transfers. Our results clearly confirm that rollout consistently improves performance and robustness across various scenarios, substantiating and extending the findings presented by Bertsekas (2024). Another appealing aspect of MELRO is its capacity for adaptive online regularization; regularizing the lookahead reward during online planning allows for the on-the-fly modification of the entropy weight, thereby significantly facilitating and enhancing the computational efficiency of identifying the optimal regularization trade-off in comparison to conventional MaxEntRL and domain randomization. We believe that our work constitutes an important contribution to the research and deployment of autonomous agents that are robust to the unexpected.

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

## A  MELRO in Detail

In Algorithm 1 and in Algorithm 2, we provide a detailed description of the off-line training and on-line play of MELRO.

---

**Algorithm 1:** MELRO (off-line training)

---

**input** : $\theta_s, \theta_r, \theta_q, \psi, \chi$: initial transition, reward, inference, policy and critic parameters

$\quad\quad\quad\quad\omega, \gamma$: expected return weighting factor and discount factor

$\quad\quad\quad\quad n_{init}, T_{init}$: number of samples, and time steps for model pre-training

$\quad\quad\quad\quad n_t, T_{train}$: number of samples, and time steps for model training

$\quad\quad\quad\quad D, \bar{D}$: optimization steps for model and for policy and critic

$\quad\quad\quad\quad H_{policy}, T_{policy}$: number of samples and time steps for policy and critic training

$\quad\quad\quad\quad \mathcal{B}$: replay buffer

$\quad\quad\quad\quad env_{train}$: unperturbed training environment.

**output:** $p_{\theta_s}, r_{\theta_r}$ and $q_{\theta_q}$: learned transition, reward and inference functions

$\quad\quad\quad\quad\pi_\psi, v_\chi$: learned neural base policy and critic

**1 begin**

**2** $\quad$ // Pre-train model:

**3** $\quad$ Sample $n_{init}$ trajectories $\{\Gamma_{init}\}_{1:n_{init}}$ each with length $T_{init}$ from $env_{train}$ and a random policy

**4** $\quad$ $\mathcal{B} = \mathcal{B} \cup \{\Gamma_{init}\}_{1:n_{init}}$ // Add initial trajectories to buffer

**5** $\quad$ **while** *not converged* **do**

**6** $\quad\quad$ // Train model:

**7** $\quad\quad$ $\{\Gamma\}_{1:H_{\mathrm{model}}} \sim \mathcal{B}$ // Sample H trajectories from buffer

**8** $\quad\quad$ **for** $d = 1, \dots, D$ **do**

**9** $\quad\quad\quad$ Update model parameters using the ELBO and Adam

**10** $\quad\quad$ **end**

**11** $\quad\quad$ // Update policy and critic:

**12** $\quad\quad$ **for** $\bar{d} = 1, \dots, \bar{D}$ **do**

**13** $\quad\quad\quad$ Sample $H_{policy}$ trajectories $\{\Gamma\}_{1:H_{policy}}$ each with length $T_{policy}$ from $\hat{p}_\theta$ and $\hat{r}_\theta$ using $\pi_\theta$

**14** $\quad\quad\quad$ Compute TD($\lambda$) returns $R$ using Equation 11 $\forall t \in \{1 : T_{policy}\}$ and $\forall h \in \{1 : H_{policy}\}$

**15** $\quad\quad\quad$ Update policy and critic parameters using Equation 14 and Adam

**16** $\quad\quad$ **end**

**17** $\quad\quad$ Sample $n_t$ trajectories $\{\Gamma\}_{1:n_t}$ each with length $T_{train}$ from $env_{train}$ and $\pi_\psi$

**18** $\quad\quad$ $\mathcal{B} = \mathcal{B} \cup \{\Gamma\}_{1:n_t}$ // Add trajectories to buffer

**19** $\quad$ **end**

**20 end**

---

---

**Algorithm 2:** MELRO (on-line play)

---

**input** : $p_{\theta_s}, r_{\theta_r}$ and $q_{\theta_q}$: learned transition, reward and inference functions

$\pi_\psi, v_\chi$: learned neural base policy and critic

$\beta, D$: learning rate and optimization steps for MPC

$env_{deploy}$: perturbed deploy environment.

**output:** $(a_1, ..., a_T)$: actions

**1 begin**

**2**     $s_0 = env_{deploy}.reset()$

**3**     **for** $t = 1, \ldots, T$ **do**

**4**        Sample initial action mean and standard deviation trajectory $\Gamma_t^0 = (\hat{a}_t^0, \hat{a}_{t+1}^0, ..., \hat{a}_{t+H}^0)$ from $\pi_\psi$ and $p_{\theta_s}$

**5**        Add small uniform noise to $\Gamma_t^0$.

**6**        **for** $d = 1, \ldots, D$ **do**

**7**           Compute search direction $\Delta_{\Gamma_t^d}$ based on Equation 15 starting from $\Gamma_t^{d-1}$ using ARS, $r_{\theta_s}$ and $v_\chi$

**8**           Update action distribution parameter trajectory $\Gamma_t^d$ in the direction $\Delta_{\Gamma_t^d}$ using Adam

**9**        **end**

**10**        $a_t \sim \mathcal{N}((\mu, \sigma) = \hat{a}_t)$ `// Sample action`

**11**        $s_{t+1} = env_{deploy}.step(a_t)$

**12**     **end**

**13 end**

---

# B Hyperparameter

In this section, we list the hyperparameter search space and the optimal parameters found for all the methods used. Note that the entropy_weight for the base policy for the policy parameters, as well as for rollout, was fixed to zero. Additionally, the exploration_noise, which describes the standard deviation of a zero-centered Gaussian noise, has been added to the actions to enable exploration only in the non-entropy-regularized MBRL setting.

Table 1: MBRL Hyperparameter

| | Search Space | Maze | Simulation | Franka |
|---|---|---|---|---|
| **Critic Parameter** | | | | |
| max_iter | 1000 | | | |
| activation | [softsign, elu] | softsign | softsign | elu |
| n_hidden | [128, 256, 512, 1024] | 256 | 512 | 1024 |
| n_layers | 2 | | | |
| **Policy Parameter** | | | | |
| exploration_noise | [0.0, 0.05, 0.1] | 0.1 | 0.0 | 0.01 |
| step_size | [3e-4, 1e-4] | 3e-4 | 3e-4 | 3e-4 |
| max_iter | [20, 40, 60, 100, 120] | 60 | 100 | 60 |
| activation | softsign | | | |
| n_hidden | [64, 128, 256, 512] | 256 | 64 | 64 |
| n_layers | [2, 3] | 2 | 2 | 3 |
| discount | [0.95, 0.975, 0.99, 0.995] | 0.99 | 0.975 | 0.95 |
| entropy_weight | [0.0, 0.0001, 0.005, 0.01, 0.05, 0.1] | 3[2] | 0.01 | 0.0001 |
| n_samples | [64, 128, 256] | 64 | 128 | 64 |
| n_time_steps | [4, 6, 8, 16, 24, 32] | 10 | 6 | 6 |
| td_lambda | [0.9, 0.95, 0.99] | 0.95 | 0.95 | 0.99 |
| max_grad_norm | [1, 10, 100] | 100 | 10 | 100 |
| **Model Parameter** | | | | |
| batch_size | [32, 64, 128, 256] | 256 | 32 | 32 |
| emission_scale | [0.01, 0.05, 0.1, 0.2] | 0.2 | 0.01 | 0.01 |
| max_grad_norm | [10, 100, 1000] | 100 | 100 | 100 |
| max_iter | [10, 20, 40, 100] | 10 | 40 | 100 |
| n_time_steps | [4, 6, 8, 10] | 4 | 4 | 10 |
| step_size | [3e-3, 1e-3, 3e-4, 1e-4] | 3e-4 | 3e-4 | 3e-4 |
| initial_iter | [100, 250, 500, 1000] | 1000 | 100 | 250 |
| rnn_activation | softsign | | | |
| rnn_n_hidden | [24, 64, 128] | 24 | 24 | 128 |
| mlp_activation | softsign | | | |
| mlp_n_hidden | [128, 256, 512] | 256 | 256 | 128 |
| mlp_n_layers | [2, 3] | 2 | 3 | 3 |
| **MBRL Parameter** | | | | |
| action_repeat | 2 | | | |
| conservativity | [0.01, 0.05, 0.1] | 0.05 | 0.05 | 0.05 |
| critic_weight | [0.05, 0.1, 0.25, 0.5, 1.0] | 0.05 | 1 | 0.25 |
| n_iter_per_eval | 1e6 | | | |
| n_steps_per_eval | 10000 | | | |
| max_env_steps | 500_000 | | | |
| n_collect_steps | [10, 20, 40, 100] | 100 | 40 | 100 |
| n_initial_time_steps | [100, 500, 1000, 2500] | 100 | 1000 | 1000 |

---

[2]We used the following search space for the entropy_weight for the maze [1, 2, 3, 4, 5, 6]

Table 2: Rollout Hyperparameter Search Space

| Parameter | Values |
|---|---|
| entropy_weight | [0.0, 0.0001, 0.0005, 0.001, 0.005, 0.01, 0.05, 0.1, 0.5, 1] |
| n_base_policy_steps | [0, 1, 2, 4, 6, 8] |
| ars_n_perturbations | [8, 12, 16, 24] |
| ars_std | [0.05, 0.1, 0.15, 0.2, 0.25, 0.3] |
| ars_top_k | [2, 4, 8] |
| discount | [0.8, 0.9, 0.95, 0.999] |
| n_optimize_steps | [1, 2, 5, 10, 15, 20, 35, 45] |
| n_parallel_rollouts | 1 |
| n_planning_steps | [1, 2, 4, 6, 8, 12, 20] |
| optimizer | Adam |
| step_size | [0.001, 0.003, 0.01, 0.03, 0.1, 0.125, 0.15, 0.175, 0.2, 0.25] |

## C   Analysis of the computational costs

Although the many benefits of rollout are evident, performing online optimization can quickly become computationally expensive. In order to run MELRO at 10 Hz, as stated in Section 4.3, we implemented all components of MELRO in JAX. We further compile all computationally intensive functions, such as all world model functions, online optimization, and base policy calls, using JAX's `jit` functionality. Additionally, we deployed MELRO for inference on a MIG 1g.10gb partition of an NVIDIA A100 GPU (compare to NVIDIA (2025). This not only makes inference of the neural components faster, but also enables ARS to be completely parallelizable. As a result, the computational time of MELRO scales only linearly with the number of planning steps, the number of optimization steps, and the number of base policy steps, as visualized in Figure 15. By keeping those parameters as small as possible, MELRO can operate reliably at 10 Hz, as shown in Table 3. The same table also shows that entropy regularization during planning increases the inference time only marginally, and that the MaxEnt Base Policy can still run around ten times faster than MELRO.

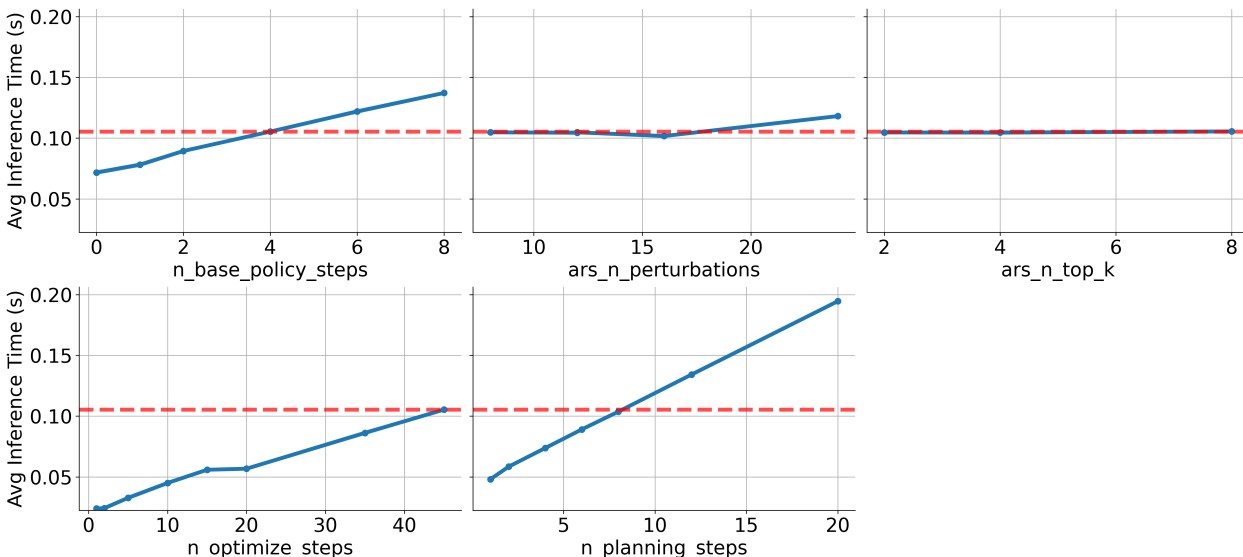

Figure 15: **Computational Costs MELRO.** Average inference times of MELRO for different hyperparameters computed during the evaluation of the Franka experiments. The red dotted line shows the mean inference time for the optimal hyperparameters used for the Franka experiments.

Table 3: **Inference time MELRO.** Statistics of the inference time in seconds of the MaxEnt base policy, MELRO without entropy regularization, and MELRO using the hyperparameters used for the Franka experiments.

| Method | average | min | max | standard deviation |
|---|---|---|---|---|
| MaxEnt Base Policy | 0.00955 | 0.008731 | 0.013814 | 0.000678 |
| MELRO w/o entropy | 0.105343 | 0.104021 | 0.109615 | 0.000668 |
| MELRO (ours) | 0.105479 | 0.10413 | 0.109755 | 0.000667 |

## D    Sensitivity Analysis

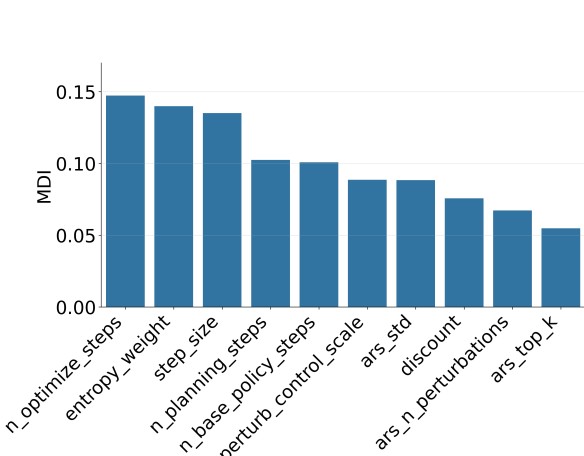

Figure 16: **Importance analysis rollout parameters.** Mean Decrease Impurity (MDI) of all rollout features for all experiments. Higher value translates to higher importance.

Figure 17: **Sensitivity analysis entropy weight in rollout.** Ratios of the entropy regularization weight of the top-5% experiments of each evaluated environment.

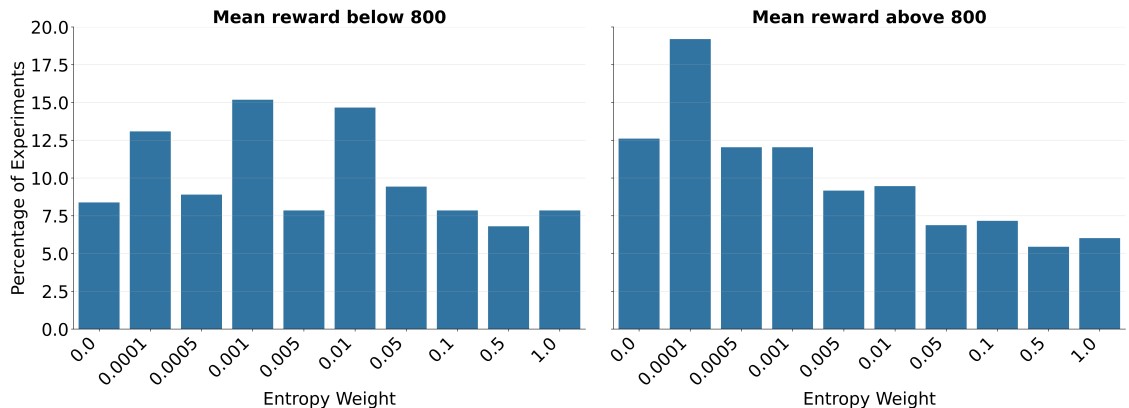

Figure 18: **Sensitivity analysis entropy weight in rollout.** Ratios of the entropy regularization weight of the top-5% experiments per perturbations divided into groups with a mean reward below and above 800.

To investigate the role of maximum-entropy regularization during planning, we first trained a random forest model on the complete set of conducted experiments, using the rollout parameters as input features and the final reward as the prediction target. Feature importance was then quantified using the Mean Decrease Impurity (MDI) metric (Breiman, 2002), with results presented in Figure 16. In a complementary analysis, we examined the distribution of entropy weights among the top-5% performing experiments for each perturbation and environment, with the resulting ratios shown in Figure 17. In order to differentiate between entropy regularization levels depending on the severity of the perturbation, we separated the perturbations based on the mean reward of the top-5% experiments into two groups: one with a mean reward above 800 and one with a mean reward below 800. The results are shown in Figure 18. Together, these evaluations highlight that entropy weight is the second most influential rollout hyperparameter, that moderate entropy regularization consistently improves planning performance, but also that excessive regularization degrades performance by inducing overly stochastic action distributions. Furthermore, in the case of less severe perturbations, small entropy regularization is more beneficial. Conversely, for more severe perturbations, greater entropic actions are more effective.

# E   Franka Panda Robot Simulation Details

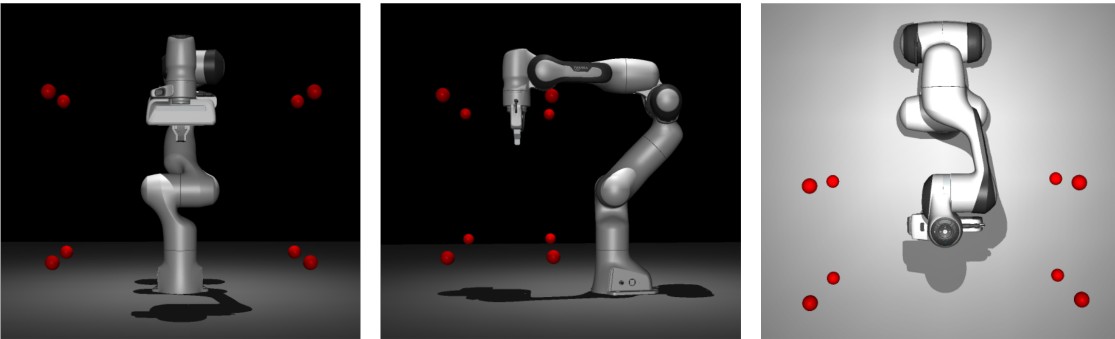

Figure 19: **Franka reach goal region.** Visualization of the goal range in the Franka Reach MuJoCo simulation. The red dots mark the corners of the cube where the goals are sampled from.

As mentioned in the main text, the Franka Panda MuJoCo Menagerie model (Zakka et al., 2022) has been used to train the methods in simulation. The desired goal position $g$ has been sampled for each trajectory uniformly from a cube relative to the base of the robot arm:

$$g \sim \mathcal{U}\left[\begin{pmatrix} 0.2 \\ -0.35 \\ 0.15 \end{pmatrix}, \begin{pmatrix} 0.5 \\ 0.35 \\ 0.6 \end{pmatrix}\right]. \tag{16}$$

The borders of the box are defined in meters. Figure 19 visualizes the goal cube in the simulation. Besides the goal position, a fixed straight downward-facing quaternion $g_q = (0, 1, 0, 0)$ has been used to penalize any other end-effector rotations. The reward for an end-effector position $s$ and rotation $s_q$ has then been computed as follows:

$$r(s, s_q, g, g_q) = -\sum_{i=1}^{3} (s_i - g_i)^2 - min\{|s_q - g_q|, |s_q + g_q|\}, \tag{17}$$

using the antipodal symmetry of quaternions.

## F  Additional Task Visualizations

Figure 20 provides some additional task visualization of different perturbation types and values for the Real World Reinforcement Learning benchmark.

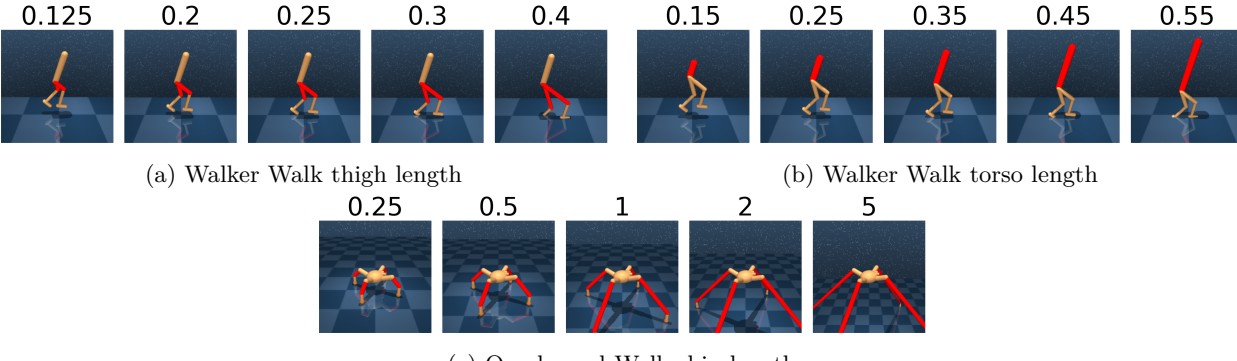

(a) Walker Walk thigh length   (b) Walker Walk torso length

(c) Quadruped Walk shin length

Figure 20: **RWRL tasks.** Visualization of some RWRL perturbations.

## G Additional Experiments Results

### G.1 Toy Example

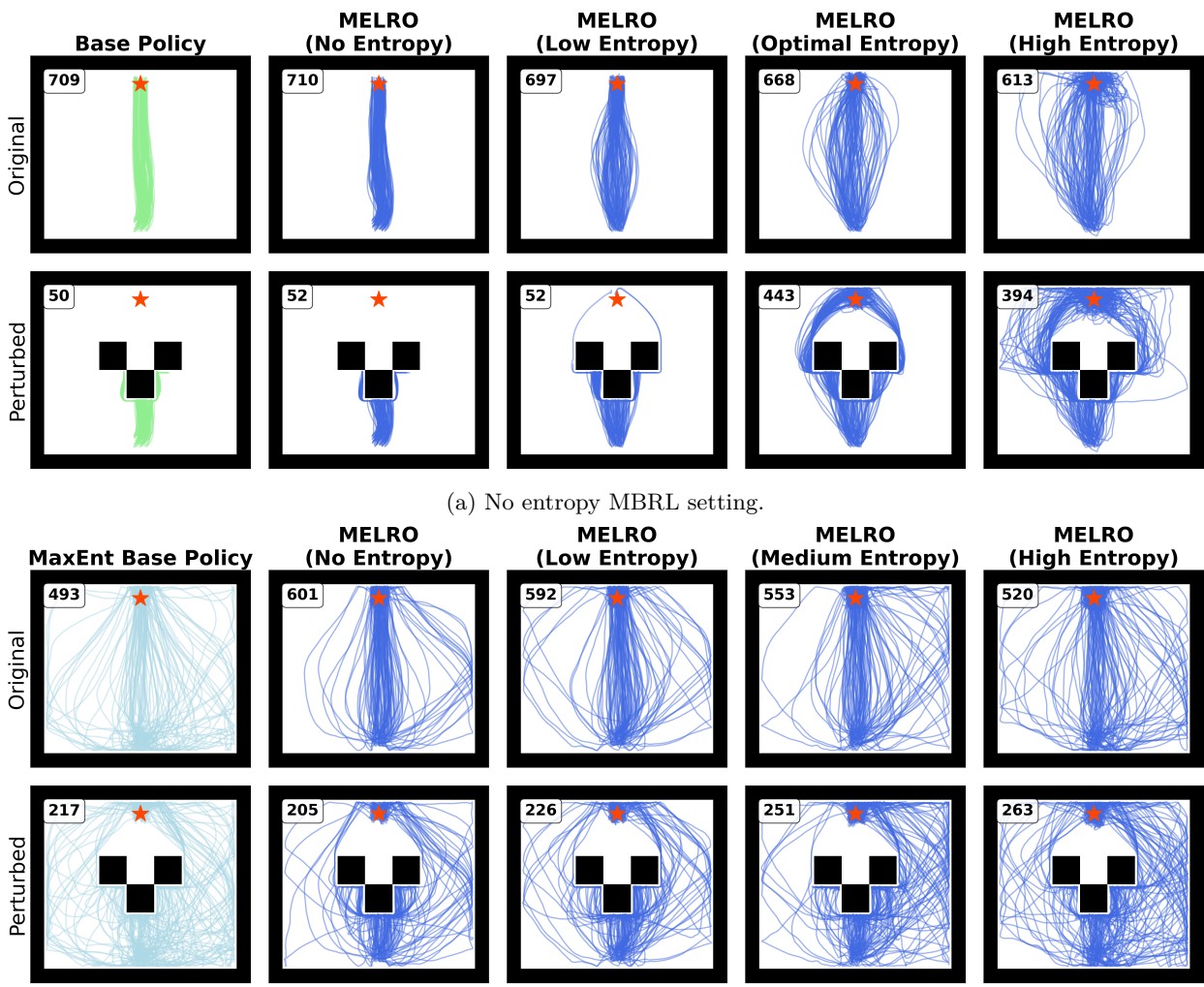

(a) No entropy MBRL setting.

(b) MaxEntRL MBRL setting.

Figure 21: **Entropy regularization toy maze.** 100 evaluation trajectories of the no entropy base policy, MaxEnt base policy, and MELRO with different entropy regularization weights for the unperturbed training maze and the perturbed evaluation maze. The components of MELRO have been trained here in the standard MBRL setting in (a) and in a maximum-entropy regularized MBRL setting in (b). The mean reward for each method and environment is shown in the top left corner of every maze. The goal is visualized by the red star.

**Impact of Entropy Regularization Weights for Planning** Figure 21 compares the trajectories generated by MELRO with different entropy regularization weights during planning. The components were trained in two frameworks: a standard MBRL framework (Figure 21a) and a maximum-entropy-regularized MBRL framework (Figure 21b). In addition to the results presented in Section 4.1, these findings demonstrate that MELRO enhances performance in low as well as in highly maximum-entropy-regularized MBRL settings. Furthermore, the trajectories demonstrate that MELRO achieves superior outcomes in the original training maze, even without or with low entropy regularization during planning. This finding indicates that

rollout itself can improve performance for specific perturbations if the model, base policy, and critic have been trained using MaxEntRL.

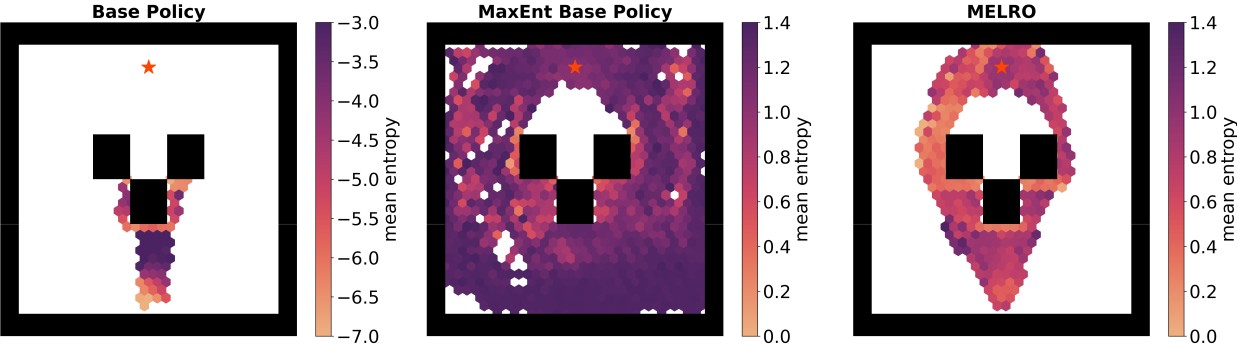

Figure 22: **Entropy visualization toy maze.** Hexbin heatmaps showing the mean entropy values at different positions in the toy maze of 100 evaluation trajectories of the low entropy base policy, MaxEnt base policy, and MELRO. The goal is visualized by the red star.

**Entropy Visualization**   As shown by (Eysenbach & Levine, 2022), maximum-entropy regularization enables dynamic adaptation of the action noise level according to the current state. This enables the policy to reduce the entropy level, provided that the reward can be increased accordingly. As illustrated in Figure 22, this phenomenon is also evident in the entropy levels of the low entropy base policy in the toy maze. At the beginning of each episode and when the point mass collides with the blocks in the middle of the maze, the entropy values are small. They then increase as the velocity increases. However, this is only possible if the reward signal is sufficiently strong to compensate for the maximum-entropy regularization penalty. This is particularly challenging at the initial positions of the maze, due to the minimal change in reward resulting from the exponential reward function. Evidence of this phenomenon is apparent in the entropy heatmap of the MaxEnt base policy in Figure 22. At the start of each episode, the MaxEnt base policy consistently generates high entropy values, causing the pointmass to be steered in all directions. It is only when the pointmass is closer to the goal that the MaxEnt base policy adapts the entropy values.

As illustrated in Figure 22, MELRO enhances entropy regularization in two distinct ways. Firstly, the overall entropy scale is increased to match the scale of the MaxEnt base policy, which introduces more diverse paths and ultimately enables the pointmass to overcome obstacles. Secondly, despite the overall increase in entropy, MELRO can maintain dynamic entropy adaptation in a manner analogous to the low entropy base policy. This enables MELRO to steer the pointmass with greater precision towards the target, particularly at significant states such as the beginning of each episode, in the proximity of the blocks, and after passing them.

### G.2 Simulation

In this chapter, we provide the complete results for all simulated experiments.

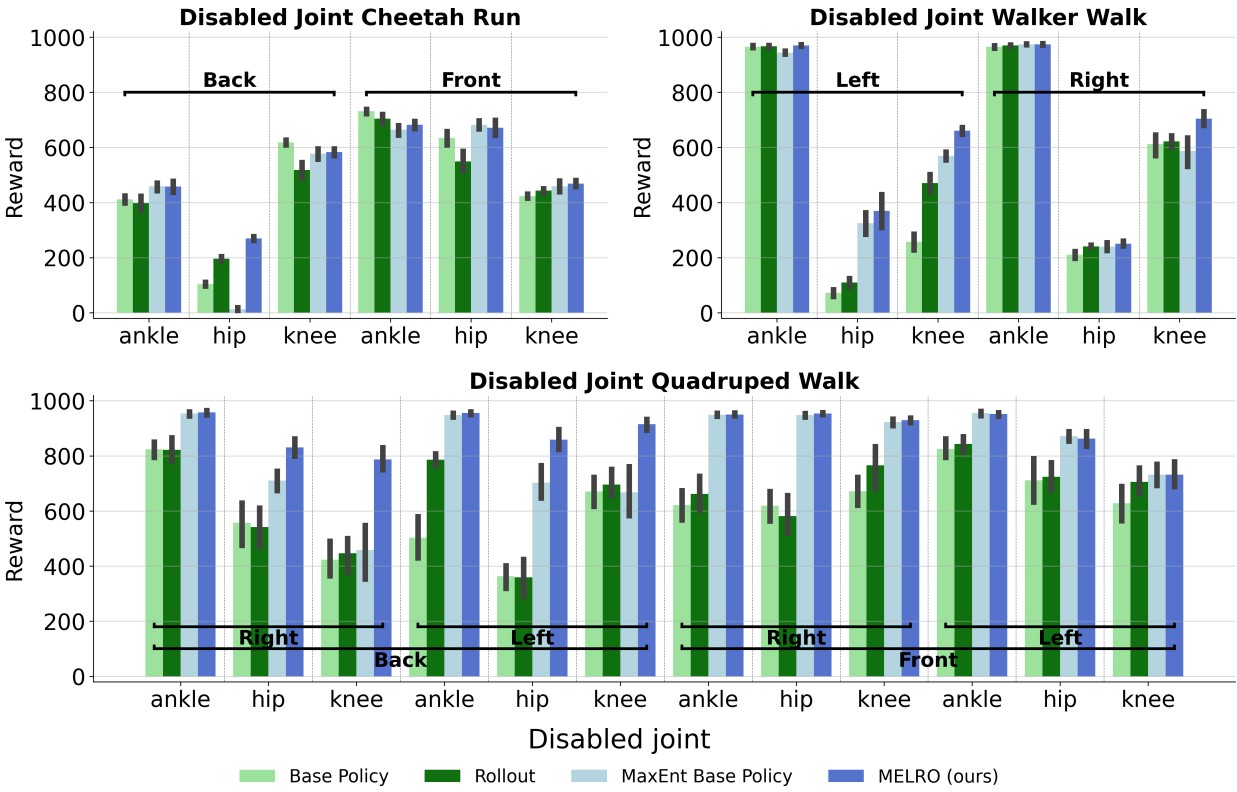

Figure 23: **Disabled joints results.** Average reward and 95%-CI for all disabled joint perturbations. The horizontal brackets indicate the positioning of the joints.

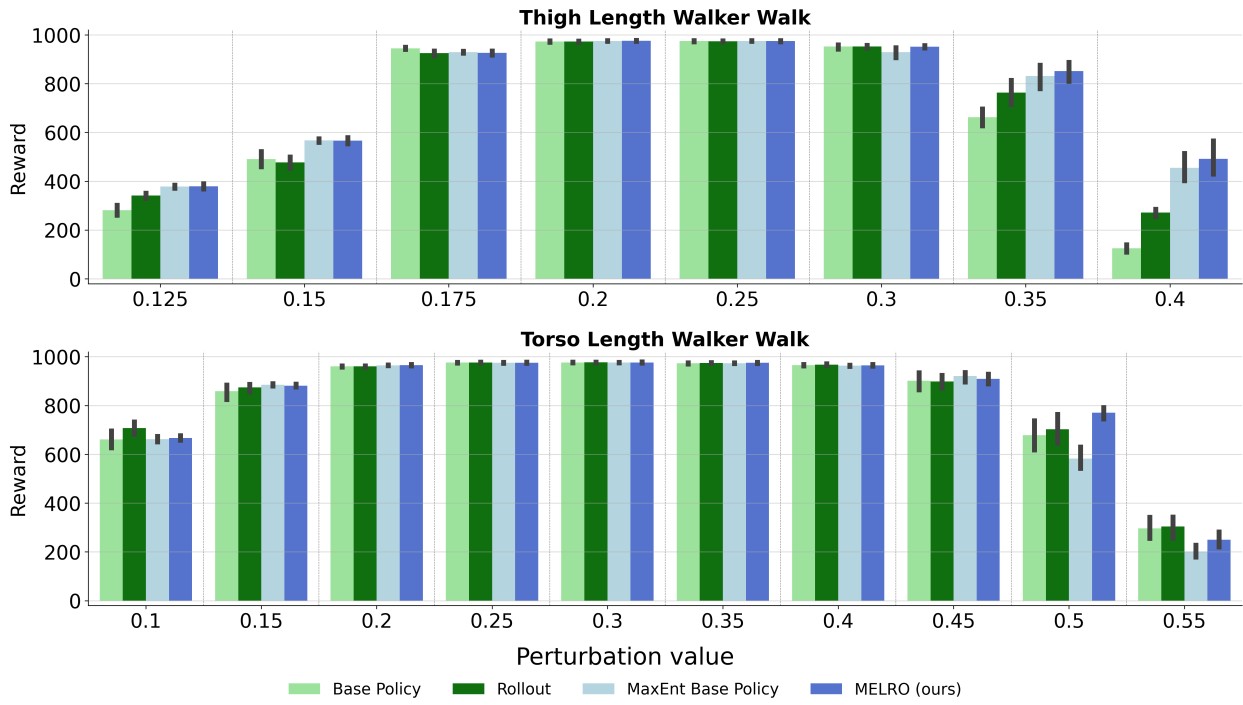

Figure 24: **Walker RWRL perturbations results.** Average reward and 95%-CI for all Walker RWRL perturbations.

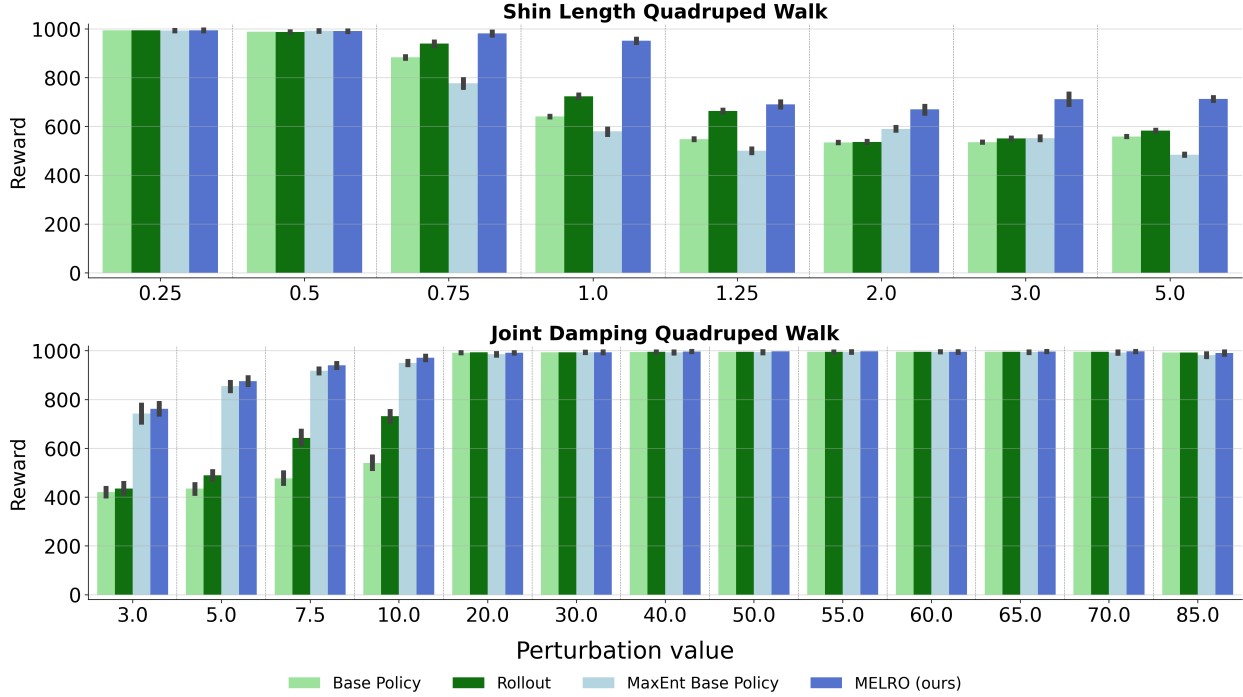

Figure 25: **Quadruped RWRL perturbations results.** Average reward and 95%-CI for all Quadruped RWRL perturbations.

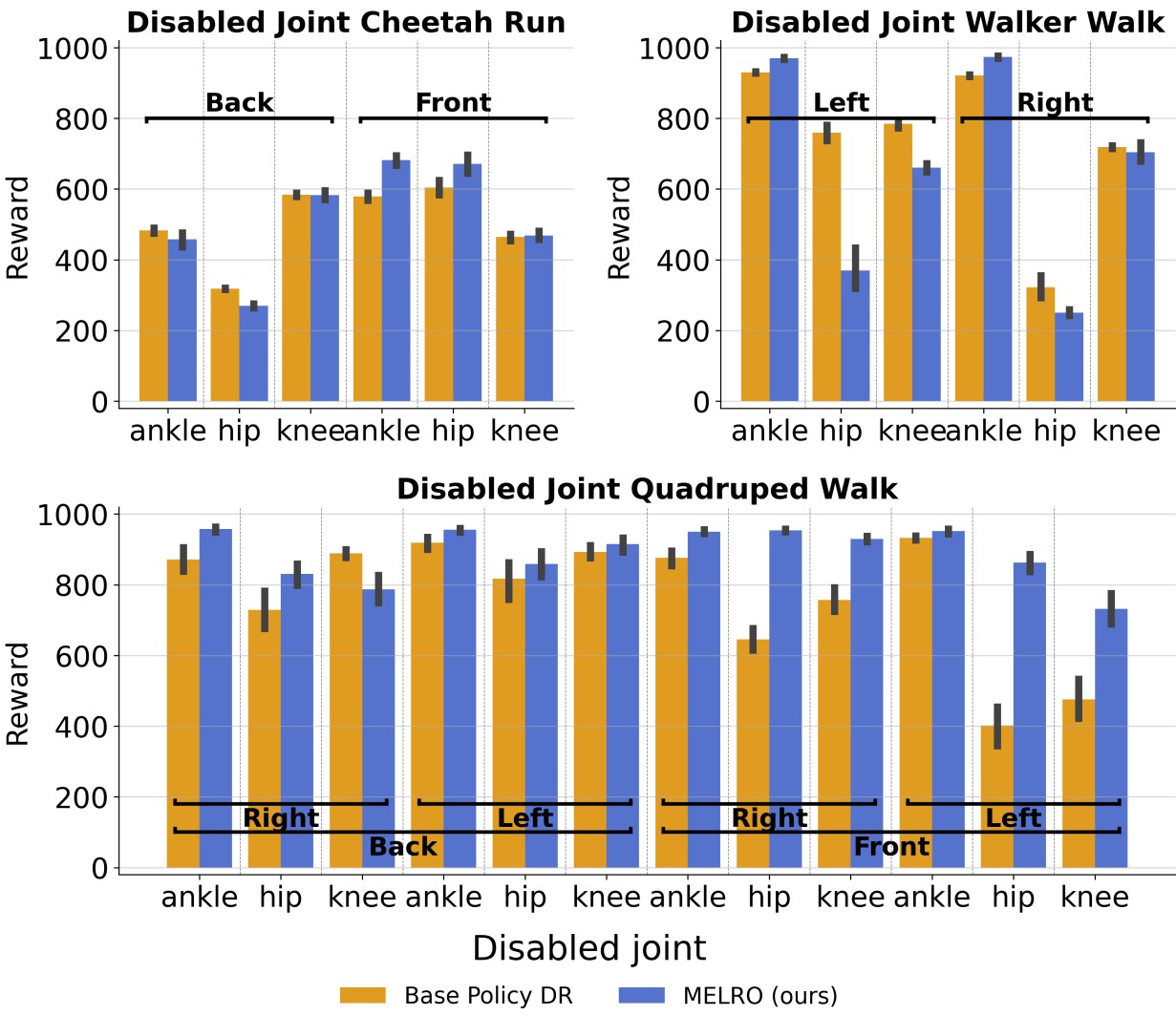

Figure 26: **Disabled joints results.** Average reward and 95%-CI for all disabled joint perturbations. The horizontal brackets indicate the positioning of the joints.

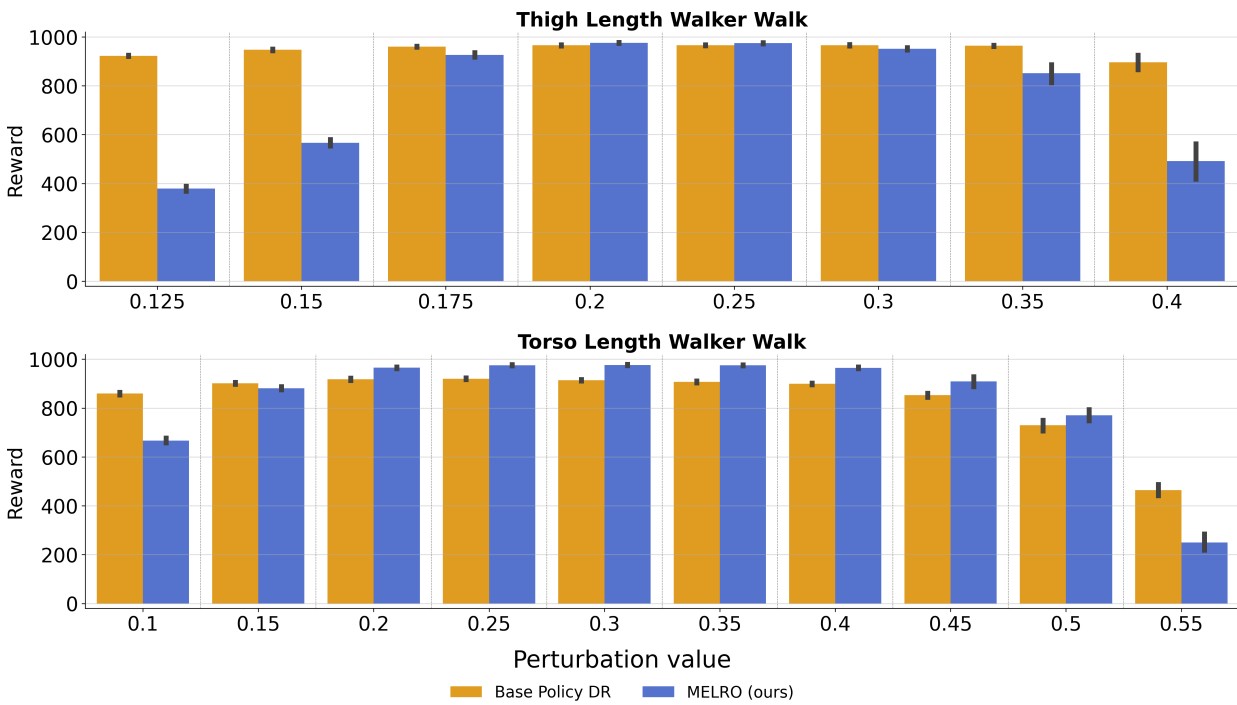

Figure 27: **Walker RWRL perturbations results.** Average reward and 95%-CI for all Walker RWRL perturbations.

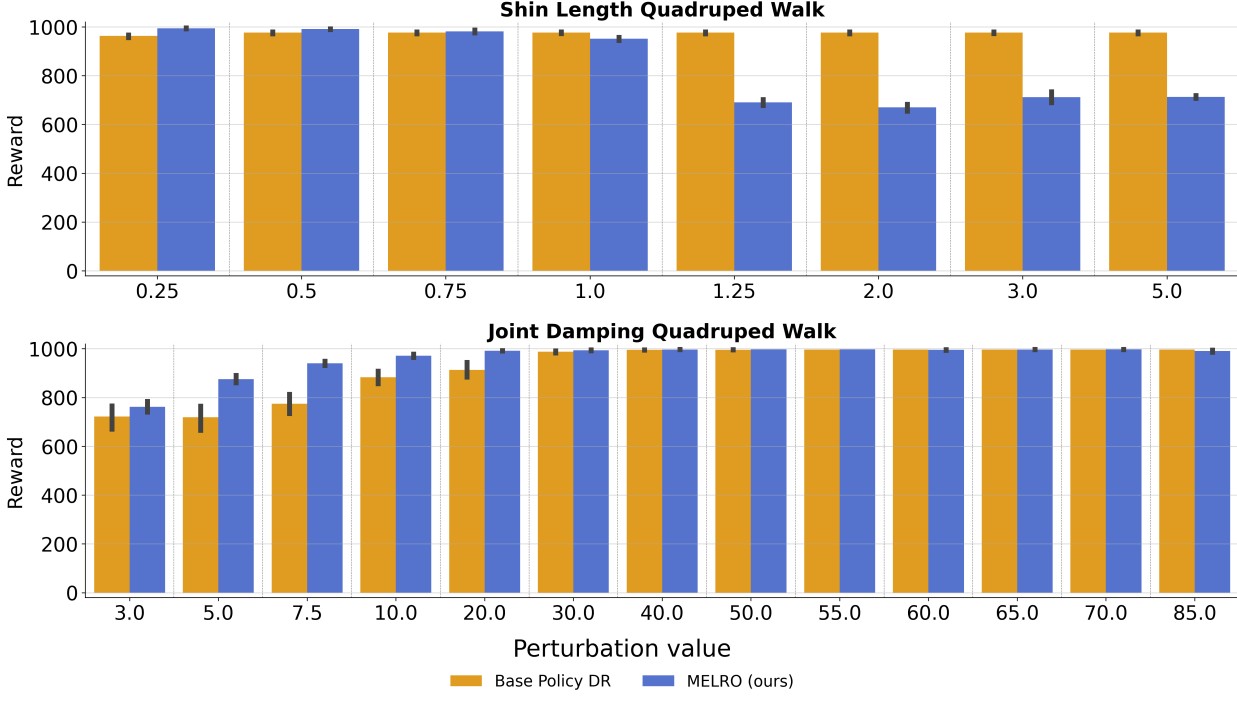

Figure 28: **Quadruped RWRL perturbations results.** Average reward and 95%-CI for all Quadruped RWRL perturbations.

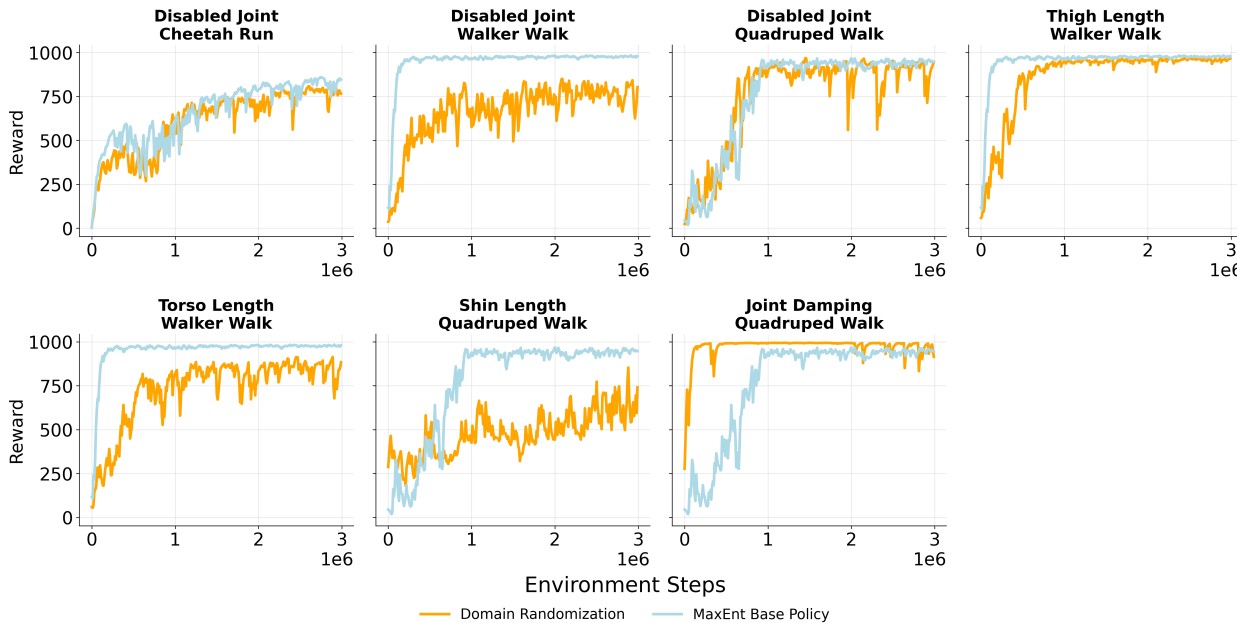

Figure 29: **MaxEntRL vs. domain randomization training curves.** Reward training curves for the base policies trained using MaxEntRL and domain randomization for all evaluated tasks individually.

