# OpenReview forum: "Inherently Robust Control through Maximum-Entropy Learning-Based Rollout"
_TMLR — Accepted by TMLR_

### Review · Reviewer_uZk7 · 2025-08-20

**Summary Of Contributions:**

This paper proposed a sim-to-real method called MELRO complementary to the domain randomization methods. MELRO is claimed to be effective when domain randomization methods are too costly to exhaust all the parameters MELRO incorporates rollout online planning to the maximum entropy framework, replacing the robustness against environments brought by domain randomization with 1) maximum entropy robustness and 2) online replanning for residual uncertainties. On simulation and a real Franka robot arm the authors tested the utilities of MELRO.

**Audience:**

Yes

**Audience Explanation:**

While I believe the paper is not ready for publication, I think the idea of "approximating the robustness of domain randomization for better sim2real with a computationally light method" is interesting and useful. The method proposed by the authors is potentially one of them, but I do think significant expansion is required for the proposed MERLO to be seriously considered effective.

**Claims And Evidence:**

No

**Claims Explanation:**

The main motivation of the paper lies in alleviating the burden of domain randomization yet keeping the robustness against environmental shift by the proposed MERLO. However, it is quite unclear to me that how or how much that is possible. The authors refer to the maximum entropy framework as "inherently robust". But how robust is that compared to domain randomization, especially when compared to DR that explicitly treats the shift as an instance of the env distribution? Does the MaxEnt framework, when formulating the entropy in Eq. (2), incorporates any idea related to such env shift? If the entropy itself contains no other information than state/actions, then why would it have the potential for sim2real and be a good alternative to DR? These concerns are not touched upon in the paper. While the online part of MERLO could be helpful by generating imaginary rollouts according to the world model, its answer to the aforementioned question is still unknown.

Empirically, the current experiments lack also evidence on 1) how MERLO is efficient and performing when compared to existing DR methods 2) the role played by MaxEnt and rollout alone.  While Figure 4 shows an ablation study, its scope is limited to the MERLO components, and in some cases MERLO shows almost the same performance as the MaxEnt base policy.

**Requested Changes:**

Currently, the paper has only 9 pages. This is significantly shorter than even the regular paper limit (12 pages) of TMLR. I believe this room can be utilized to do at least the following:
- more explanations on why the maximum entropy framework would work in the offline setting, why it has the desired robustness to help a smooth sim2real transfer. If the vanilla MaxEnt does not work, then what extensions are needed? For example, trajectory-wise or even env-wise entropy?
- more experimenst comparing against DR methods. It is necessary to show how much performance loss and computational loss are incurred
- more explanations on the framework itself: there are many components in MERLO like world model, recurrent net, etc. Why these components are chosen? What would happen if any of them is replaced?

---

> ### Author Response · Authors · 2025-10-08
> **Reply to Review from Reviewer uZk7**
>
> Dear Reviewer uZk7,
>
> We really appreciate your feedback and believe it has helped us significantly improve the paper. Thank you very much.
>
> Additionally, we are very happy that you found the idea of our paper interesting and useful.
>
> We have uploaded a revision today, in which we have addressed the requested changes as outlined below.
>
> ## 1. Request
> > More explanations on why the maximum entropy framework would work in the offline setting, why it has the desired robustness to help a smooth sim2real transfer. If the vanilla MaxEnt does not work, then what extensions are needed? For example, trajectory-wise or even env-wise entropy?
>
> In Section 4.1 and Appendix G.1 of the revised version, we have added new experiments and evaluations on an illustrative toy example to explain why entropy-regularized planning in MELRO increases robustness and improves the success of ``sim to real'' transfers.
> The evaluations also offer a more in-depth insight into how MELRO operates, its advantages, and how it differs from vanilla MaxEntRL and domain randomization.
>
> ## 2. Request
> > More experiments comparing against DR methods. It is necessary to show how much performance loss and computational loss are incurred.
>
> We have trained additional policies using domain randomization for the toy maze example and the simulation experiments.
> The results and comparisons with MELRO can be found in Section 4.1 and Section 4.2 of the revised paper, as well as in Appendix G.2.
> We also conducted a more detailed analysis of the various strategies generated by MELRO and domain randomization for the toy maze.
> Additionally, we have added the training reward curves for both frameworks to quantify the computational loss (compare to Figures 4 and 11).
>
> ## 3. Request
> > More explanations on the framework itself: there are many components in MERLO like world model, recurrent net, etc. Why these components are chosen? What would happen if any of them is replaced?
>
> The fundamental concept of MELRO, combining MaxEntRL and rollout, is agnostic to most of the specifications of the selected components.
> However, we provided a more thorough description of the components in the revised version of Section 3 and also removed explanations of components that are actually already used in Bayer (2021).

---

### Review · Reviewer_fcZj · 2025-08-21

**Summary Of Contributions:**

This paper proposes to modify planning methods (à la Model Predictive Control) to include maximum entropy optimizing _during planning_, not just in the proposal policy itself (as MaxEntRL would do). Extending the intuition of MaxEntRL, this pushes agents to not just learn "myopically" robust policies, but to also robustly plan. The authors argue that this robustness is an important ingredient in _sim to real_ transfer, and demonstrate improved performance on a variety of perturbed tasks, and an actual sim to real transfer.

**Additional Comments:**

Minor: Maybe it's just my perspective on this, but I'm surprised to not see MPC mentioned by name. Is it no longer called that?

**Audience:**

Yes

**Audience Explanation:**

The sim2real problem is an important one, and this is an interesting addition to the field of planning methods that combines ingredients known to work in other domains.

**Claims And Evidence:**

Yes

**Claims Explanation:**

The authors claim that MELRO improves generalization performance in simulation and transfer, which are both backed by evidence. The authors also perform experiments to better understand where the performance increase comes from.

**Requested Changes:**

I think the main thing I find lacking in the paper is the ablation & introspection part. Section 4.1.2 could easily be 2 pages & have way more information than what's in Figures 5 (the only non-reward-based plot of the paper!). There's a lot to unpack in maximum entropy methods and the effect they have on action choices, but none of that is explored here, and so we're left with only hints of why planning maxent matters.

Relatedly, the appendix seems to list sets of possible hyperparameters, but where are the results of these? As is, I'm not sure there's any explanation of the impact/sensitivity of those values in the paper, in particular the impact of the entropy parameters seems important to report.

I may be misunderstanding the setup here, but IIUC the Rollout baseline has no MaxEnt at all. Isn't a missing baseline one where the policy  is _trained_ with the MELRO setup, but where the entropy term is turned off at inference? This would further isolate the impact of entropy in planning.

---

> ### Author Response · Authors · 2025-10-08
> **Reply to Review from Reviewer fcZj**
>
> Dear Reviewer fcZj,
>
> First, we would like to thank you for the constructive and helpful feedback. We are also happy that you found our paper interesting and that you consider the experiments to support our findings.
>
> We have addressed the requested changes in the revision uploaded today, as outlined below.
>
> ## 1. Request
> > I think the main thing I find lacking in the paper is the ablation & introspection part. Section 4.1.2 could easily be 2 pages & have way more information than what's in Figures 5 (the only non-reward-based plot of the paper!). There's a lot to unpack in maximum entropy methods and their effect they have on action choices, but none of that is explored here, and so we're left with only hints of why planning maxent matters.
>
> In the revised version, we have added new results from multiple experiments and analyses conducted on a toy maze problem to Section 4.1 and Appendix G.1.
> This addition aims to highlight the differences between the strategies generated by traditional MaxEntRL, domain randomization, and MELRO.
> They also dive deeper into the effects of maximum entropy regularization for planning in general.
>
>
> ## 2. Request
> > Relatedly, the Appendix seems to list sets of possible hyperparameters, but where are the results of these? As is, I'm not sure there's any explanation of the impact/sensitivity of those values in the paper, in particular, the impact of the entropy parameters seems important to report.
>
> We tackled this request in two ways:
> First, we have extended the table listing the hyperparameter search space for the MBRL phase in Appendix B with the final hyperparameter obtained through random search.
> Second, to demonstrate the importance of entropy regularization weights, we conducted an importance analysis using the Mean Decrease Impurity metric and evaluated the specific weights distribution for the top-5% performing experiments.
> These findings can be found in Appendix D.
>
> ## 3. Request
> > I may be misunderstanding the setup here, but IIUC the Rollout baseline has no MaxEnt at all. Isn't a missing baseline one where the policy is trained with the MELRO setup, but where the entropy term is turned off at inference? This would further isolate the impact of entropy in planning.
>
> We reran the HPS for the rollout parameters of the simulation experiments, adding the option to turn off the entropy weight during planning.
> The baseline mentioned in the request is now part of the HPS, and the concrete question about the impact of entropy during planning is answered in the sensitivity analysis in Appendix C.
> Additionally, we have included the baseline MELRO without entropy regularization for the ``sim to real'' Franka experiments.
> These results are shown in Figure 13.
>
> ## Additional Comment:
> > Minor: Maybe it's just my perspective on this, but I'm surprised to not see MPC mentioned by name. Is it no longer called that?
>
> Yes, rollout and MPC are related terms, similar to receding horizon control and limited lookahead control. We decided to use rollout as in the work of Bertsekas. To avoid confusion, we have added a footnote to Section 2.3.

---

### Review · Reviewer_c2f6 · 2025-10-14

**Summary Of Contributions:**

## Summary

This paper proposes MELRO (Maximum-Entropy Learning-Based Rollout), a control framework that couples maximum-entropy RL with online rollout planning. Offline, a variational state-space world model, a stochastic Gaussian policy, and a critic are trained with MaxEnt regularization. Online planning is performed in the space of the policy’s action-distribution parameters with an entropy-regularized objective and a truncated base-policy rollout before bootstrapping with the critic. Experiments on DMC tasks with RWRL-style and non-parametric “disabled joint” perturbations show MELRO improves over a MaxEnt base policy and over standard rollout. A Franka Panda reaching task reports lower goal distance than the MaxEnt base policy.

## Strengths

1. Methodological and design choices. Authors propose clear and principled integration of MaxEnt RL and rollout, including planning in $(\mu,\sigma)$ space and explicit entropy regularization during planning.

2. Broad empirical probe of robustness. Evaluations cover parametric (length, damping) and non-parametric (disabled joints) disturbances, with consistent gains vs. baselines and 95% CIs over 10 seeds.

3. Real-robot demo. A sim-to-real reaching task on a Franka Panda shows measurable performance gains and practical feasibility at 10 Hz command rate with a 1 kHz PD loop.

4. Clarity and presentation: The paper is overall crisp in exposition and well-organized.


## Weaknesses

1. No analysis of the computational costs. Due to the online planning stage, authors should include the computational costs, inference time, and the hardware used.

2. Lack of a domain randomization baseline. The authors mention that DR is an orthogonal direction, but it is unclear what the benefits of MELRO are compared to DR. A better explanation would help readers better contextualize the proposed method.

3. Figure 9 is unclear: which is MELRO and which is Max Ent?

4. (minor) In some tasks, base policy or rollout alone can outperform MELRO (e.g., disabled joint walker walk, torso length walker for 0.5) from Fig. 4. Why could this happen?


This said, I believe the paper is an interesting read and of interest to the TMLR community, and I would recommend it for acceptance.

**Audience:**

Yes

**Audience Explanation:**

RL and robotics are hot topics and the authors present an interesting work. This work is of interest for the TMLR community.

**Claims And Evidence:**

Yes

**Claims Explanation:**

I believe the authors did a great job supporting the paper claims by accurate, convincing and clear evidence.

**Requested Changes:**

1. Please include an analysis of the computational costs as per W1 (critical)
2. Either include an experiment or expand on the differences in performance of domain randomization and MELRO as per W2 (critical)
3. Figure 9 fix as per W3 (minor)
4. Explanation of why MELRO sometimes underperforms (minor)

---

> ### Author Response · Authors · 2025-10-24
> **Reply to Review from Reviewer c2f6**
>
> Dear Reviewer c2f6,
>
> Thank you very much for taking the time to review our paper. We sincerely appreciate your thoughtful and constructive feedback, and are happy that you found our paper crisp and interesting.
>
> We have uploaded a revised version that addresses the changes you requested, as outlined below.
>
> ## 1. Request
> > Please include an analysis of the computational costs as per W1 (critical)
>
> We conducted a comprehensive analysis of the computational inference costs of MELRO in Appendix C.
> Additionally, we described our hardware and software setup.
>
> ## 2. Request
> > Either include an experiment or expand on the differences in performance of domain randomization and MELRO as per W2 (critical)
>
> We addressed this request in two ways:
> First, we trained policies using domain randomization on all simulation experiments, and added the results to Section 4.2.
> Second, to demonstrate the difference in the generated control strategies generated by maximum-entropy regularization and domain randomization, we conducted experiments on a maze environment.
> This analysis also dives deeper into the effects of entropy-regularized planning.
> These results are given in Section 4.1 and in Appendix G.1.
>
> ## 3. Request
> > Figure 9 fix as per W3 (minor)
>
> We fixed the plot and updated the caption to be more detailed.
>
> ## 4. Request
> > Explanation of why MELRO sometimes underperforms (minor)
>
> We extended the results of the simulation experiments in Section 4.2 with a new paragraph discussing the limitations of maximum-entropy regularization.
> Additionally, we could improve the performance, especially for the ``Walker Walk'' torso length 0.5 task of rollout and MELRO through better rollout hyperparameters.

---

### Author Response · Authors · 2025-10-24
**General response to reviews**

Dear Reviewers,

We would like to thank you for your time and effort to review our paper, and sincerely appreciate your thoughtful and constructive feedback.

We have just uploaded a revised version.

While we have provided individual comments for each review regarding how we addressed the individual requests, we have also summarized the additions and changes below.

## Comparison to Domain Randomization
We trained policies using domain randomization on all simulation experiments, and have added the results to Section 4.2 and Appendix G.2.
In these Sections, we showcase that entropy-regularized planning is a real viable and competitive alternative to domain randomization, despite the clear benefit of policies trained under domain randomization in terms of the opportunity to encounter the specific validation perturbations during the MBRL phase.
For the disabled joint tasks, MELRO has even been shown to outperform policies trained using domain randomization (compare to Figures 6 and 10).
Furthermore, we quantify the additional computational costs of domain randomization in Figure 11.

## More in-depth analysis of the effects and workings of entropy-regularized planning
To demonstrate the challenges of non-regularized RL, standard MaxEntRL, and domain randomization, as well as demonstrating the advantages and the effects of entropy-regularized planning in MELRO, we conducted further experiments on an illustrative toy example in Section 4.1 and Appendix G.1.
The key finding is that MELRO can maintain a better dynamic entropy adaptation: it relinquishes only a minimal amount of reward in the unperturbed environment and attains the maximum mean reward under the perturbation.

## Performance improvement of MELRO and explanation on why entropy regularization is not as effective for some environments
We have improved MELRO's performance by adjusting the rollout parameters, particularly for the 'Walker Walk' task with a torso length of 0.5.
However, maximum-entropy regularization was still not as effective for some environments, for which we give a possible explanation in a new paragraph in Section 4.2.

## Ablation MELRO without entropy-regularized planning
We ran an ablation of MELRO without entropy during planning for the Franka experiments.
This showed that rollout and maximum entropy regularization improve performance on the ``sim to real'' task both individually and jointly (see Figure 13).
To extend those results, we reran the hyperparameter search of the rollout hyperparameter, with the new option to turn entropy regularization during rollout completely off.
We then used the new results to perform a hyperparameter sensitivity analysis in Appendix D.
This shows that the entropy weight is the second most influential rollout hyperparameter, that moderate entropy regularization consistently improves planning performance, but also that excessive regularization degrades performance by inducing overly stochastic action distributions.

## Clearer explanation of the framework
The fundamental concept of MELRO, combining MaxEntRL and rollout, is agnostic to most of the specifications of the selected components.
We provided a more thorough description of the components in the revised version of Section 3 and also removed explanations of components that are actually already used in Bayer (2021).

## Computation cost analysis
We conducted a comprehensive analysis of the computational inference costs of MELRO in Appendix C.
Additionally, we described our hardware and software setup.

## More details on the hyperparameter and their impact
We have extended the table listing the hyperparameter search space for the MBRL phase in Appendix B with the final hyperparameter obtained through random search.
To demonstrate the importance of entropy regularization weights, we conducted an importance analysis using the Mean Decrease Impurity metric and evaluated the specific weights distribution for the top-5% performing experiments.
These findings can be found in Appendix D.

## More detailed captions
We have made the figure captions clearer and more detailed.

---

### Author Response · Authors · 2025-12-18

We would like to thank the reviewers once again for their extremely constructive and helpful feedback, as well as our action editor, Razvan Pascanu, for overseeing the review process and ensuring that everything ran smoothly behind the scenes.

---

### Decision · Action_Editor_9CG2 · 2025-11-26

**Recommendation:** Accept as is

**Audience:**

Yes

**Audience Explanation:**

The work provides an alternative approach to address sim-to-real, which is an important problem in the field. Given the topic and the tools used, I believe a good chunk of the community would be interested in this work.

**Claims And Evidence:**

Yes

**Claims Explanation:**

The paper provides sufficient clarity in the description of the method and sufficient empirical evidence. In particular one initial weakness of the work was not having a proper comparison with domain randomization but this has been fixed during the rebuttal period, and in its current form I think the paper is considerably stronger. All reviewers seem to agree on this front.